# Psychological, social, and health-related factors predict risk for financial exploitation

Yi Yang[1], Katherine Hackett[2], Srikar Katta[3], Rita M. Ludwig[4], Johanna Jarcho[1], Tania Giovannetti[1], Dominic S. Fareri [5,6] & David V. Smith [1,6] ✉

People lose tens of billions of dollars a year to financial exploitation in the United States alone. Few studies have examined how preferences for trust and fairness in economic activities may contribute to risk for financial exploitation. Furthermore, few studies have examined the interaction between risk factors. In three studies, we attempt to address these gaps by surveying 1918 (Study 1 = 680, Study 2 = 305, Study 3 = 933) demographically and socioeconomically diverse participants to examine putative risk factors for self-reported financial exploitation. We focused on: (1) how trust in others and fairness preferences during economic games are associated with self-reported financial exploitation; and (2) how sociodemographic and health-related factors interact with psychosocial factors to confer risk for financial exploitation. We found participants with lower socioeconomic status and poor emotion regulation skills were at the greatest risk for financial exploitation. We also found associations between greater risk for financial exploitation and poorer physical health, more severe cognitive decline, increased persuadability, and increased insensitivity to trustworthiness cues. Our findings suggest that risk for financial exploitation is dependent upon a combination of psychosocial, sociodemographic and health factors, which may lead to interventions that protect vulnerable individuals.

Financial exploitation, defined as the illegal or improper use of an adult's funds or property for another person's profit or advantage[1], results in annual losses of billions of dollars[2]. Exploitation of older adults in particular has been recognized as a serious and growing public health problem[3,4]. Multiple putative individual risk factors for financial exploitation have been identified within and across sociodemographic, health, and psychosocial domains. These include low income, poor mental health, cognitive decline, poor deception detection, and low social support[5–11]. However, little is known about the link between financial exploitation and adults' ability to integrate social information during economic decision making, and how this relationship may interact with other identified risk factors. This is a key question to consider, because processing of social/affective information (i.e., others' trustworthiness, intentions, and non-verbal cues) during financial interactions may be a promising intervention target for inoculating vulnerable adults against exploitation attempts, such as individualized psychosocial training[12,13]. Moreover, inconsistent findings across prior studies warrant the need for a comprehensive understanding of risk for financial

exploitation that can be validated across cohorts before findings are translated into actionable interventions to reduce risk. Here, we attempt to fill these gaps by systematically examining the relationships among established and risk factors for financial exploitation across three separate community samples of adults.

The first major gap in our understanding of financial exploitation is the lack of research on social information integration during economic activities as a risk factor. This gap is particularly important given that about half of financial exploitation cases are perpetrated by strangers (51%) followed by friends and family (34%)[2], and that the mechanisms underlying susceptibility to exploitation by different types of perpetrators may differ[14]. These findings suggest that people's social information integration regarding trust in others and fairness preferences may play an important role in shaping victims' economic decisions. Economic games used in studies of behavioral economics (e.g., Ultimatum Game[15]; the Trust Game[16]) have helped advance understanding of social information integration during financial decision making in the laboratory[17–19]. The Ultimatum Game assesses

[1]Temple University, Philadelphia, PA, USA. [2]Icahn School of Medicine at Mount Sinai, New York, NY, USA. [3]Duke University, Durham, NC, USA. [4]University of Pennsylvania, Philadelphia, PA, USA. [5]Adelphi University, Garden City, NY, USA. [6]These authors contributed equally: Dominic S. Fareri, David V. Smith. ✉e-mail: david.v.smith@temple.edu

preferences for fairness by proposing fair and unfair monetary offers to participants and measuring their acceptance and rejection rates dependent on offer type. Fairness preferences have been shown to be associated with contract choice. Employees who are more concerned with fairness tend to choose a bonus contract which offers non-binding bonus payment if their performance is satisfactory[20], which may make people vulnerable to financial exploitation (e.g., accepting a performance-based financial promise that cannot be enforced by a third party and becoming exploited if the performance-based bonus is not fulfilled). The Trust Game assesses trust in others by recording the amount of money one decides to invest in another person given subjective expectation of reciprocation. Differing levels of baseline interpersonal trust may relate to individuals' likelihood of falling prey to bad-faith actors. Despite their potential association with financial exploitation risk, the Ultimatum Game and the Trust Game have rarely been studied alongside risk factors for financial exploitation.

The second major gap is in studying whether interactions among multiple factors contribute to the likelihood of financial exploitation. In the real world, individuals have a constellation of background experiences and existing attributes that affect their decision-making. We posit that noted inconsistencies in the existing literature results from studying risk factors in isolation, rather than evaluating their interaction. For example, worse physical health[21,22] and worse cognitive functioning[2,23,24] have been consistently associated with higher risk for financial exploitation. However, socioeconomic status (SES)[25,26], trust in others[27,28], emotion regulation[29], and social support[27,30] have been contradictory. Results from research on age effects[6,31] also did not show that older people are particularly more vulnerable to financial exploitation, contrary to both popular and scientific opinions. One possible explanation for these inconsistent observations of a single risk factor's relationship with financial exploitation might be its potential interaction(s) with other risk factors. While there exists an abundance of research on single factors, in contrast, only a handful of studies have touched upon interactions between two or more of them[26,32,33]. Although previous work has identified associations between risk for financial exploitation, sociodemographic factors, health, and psychosocial factors, it remains unclear whether these variables interact to predict risk for financial exploitation. Given that previous work has largely focused on single factors that contribute to risk of financial exploitation, our primary goal was to assess how risk for financial exploitation was moderated by other factors. We specifically focused on how sociodemographic factors (e.g., socioeconomic status, age, etc.) and health-related factors may moderate psychological factors such as psychosocial functionings (e.g., trust in others, fairness preference, persuadability, need to belong, and emotion regulation). Such "sociodemographics vs health" (Study 1 & 2) and "sociodemographic vs psychosocial functioning" (Study 1, 2 & 3) combinations allows for a relatively more comprehensive and holistic approach to examine financial exploitation risk factors, integrating both external (sociodemographic and health-related factors) and internal (psychological) factors. This approach acknowledges the complex, multifaceted nature of financial exploitation vulnerability.

Neglecting to examine trust in others and fairness preference during economic games as a risk factor along with interactions between different risk factors limits our ability to comprehensively understand risk for financial exploitation, with downstream limitations on developing effective prevention and intervention programs. The present research addresses these literature gaps in a series of three studies looking at risk for financial exploitation for both older (Study 1, 2 & 3) and younger adults (Study 3). Below we introduce how we quantify risk for financial exploitation and hypotheses of each study.

To quantify the risk for financial exploitation, we employed the Older Adult Financial Exploitation Measure (OAFEM[1], all three studies). OAFEM is a 30-item scale that measures key behaviors related to older adults' financial exploitation. Each item maps on one of five financial exploitation concept groups (i.e., Abuse of Trust, Financial Entitlement, Coercion, Signs of Possible Abuse, and Theft & Scams. See Table S9 for item mapping). One

of the five concept groups is "Abuse of Trust" with eight items mapped on to it. Several items that are not mapped on "Abuse of Trust" may also relate to trust in others, such as the "Theft & Scam" item "Has someone overcharged you for work or services that were done poorly or never done?" Although OAFEM concept mapping does not have a group directly related to fairness preference, we argue that several items may nevertheless relate to fairness preference. For example, for people who have answered "Yes" to the "Theft & Scam" item "Has someone taken your money to do something for you but never did?", they may have regarded it was fair to the perpetrator to use the money to do a certain thing and therefore allowed the perpetrator to take their money.

We also adopted the Short Form of Financial Exploitation Vulnerability Scale (FEVS-SF[14]) as an additional measure of risk for financial exploitation (Study 3 only). The FEVS-SF is a nine-item scale that probes contextual features of the environment in which a person is making a financial decision. Unlike the OAFEM, items in the FEVS-SF are not mapped onto specific concept groups by its developers. However, several of the items may still relate to trust and fairness preference (Table S10). For example, people who rated higher on the item "How often do you wish you had someone to talk to about financial decisions, transactions, or plans?" may be more likely to trust financial advice from other people, including unsolicited ones from scammers. For fairness preference, people who rated higher on the item "How often do you feel anxious about your financial decisions and/or transactions?" may be anxious about being treated unfairly when making financial decisions and fall prey to scammers who offer ostensibly fair deals or ostensibly unfair deals that are allegedly more advantageous to the victim.

We conducted three studies to address the gaps of (1) neglecting to examine trust in others and fairness preference during economic games as financial exploitation risk factors, and (2) neglecting to examine interactions between different risk factors, which might have contributed to inconsistent findings in the literature. First, given the focus on older adults in previous financial exploitation research[31], in Study 1 we focused on risk factors of financial exploitation within older adults from our local community in Philadelphia, PA, USA. We used the 30-item OAFEM[1] that measures key behaviors related to older adults' financial exploitation. Our primary goals were to identify (1) how trust in others and fairness preference across social context in economic activities may be associated with self-reported financial exploitation, and (2) how the association between individual socioeconomic status and self-reported financial exploitation may be moderated by health-related and psychosocial factors. In Study 2, we sought to replicate our findings from Study 1 in a more diversified population within the Pennsylvania region, USA. Building off findings of interactions between socioeconomic status and several psychosocial factors from Study 1 and Study 2, whose cohorts included only older people from the Philadelphia metropolitan area and Pennsylvania region and lacked diversity in terms of race (majority White), in Study 3 we expanded our sample's age range, racial diversity, and geographical range, and examined how sociodemographic and health-related factors may moderate the association between psychosocial factors and financial exploitation. Age and geographical location have been related to idiosyncrasies in experiences related to financial exploitation, such as prevalence and types of scams[34–36]. Therefore, given the increased sample diversity in age and geographical range, besides measuring financial exploitation related behaviors using OAFEM, we also employed the FEVS-SF[14], which focuses on contextual information of the environment in which a person is making a financial decision and allows us to detect contextual risk of financial exploitation. Using three studies and three different cohorts, we investigated a range of risk factor domains including trust in others and fairness preference during economic games, sociodemographics, health-related factors, and psychosocial factors. We believe our findings contribute to a more comprehensive and reliable understanding of financial exploitation risk, and hope that insights can be used to guide healthcare, governmental and financial industries in combating financial exploitation.

## Methods

### Data collection, procedure, and analyses for all studies

All data were collected using Qualtrics panels. The surveys were sent to participants based on their eligibility and the targeted sociodemographic brackets of each study. Data from participants who failed Qualtrics' internal quality check (e.g., anyone completing in less than half of the median completion time of the "soft launch" during which 10% of the target sample size was collected for Qualtrics' reviewing purposes) and attention checks (e.g., asking participants to "select the 'strongly disagree'") were excluded from a study directly by Qualtrics prior to data transfer. In each specific analysis we also excluded participants whose data needed for the analysis was not collected due to Qualtrics technical error (see Tables S1–S8 for sample size and measure summary of each analysis). We aimed at an even split on certain demographic factors and discussed our goals and criteria when setting up each panel study with Qualtrics. However, due to real-life constraints (e.g., difficulty in recruiting non-White racial minorities), Qualtrics was only able to recruit samples as we reported in the manuscript. Qualtrics was not able to achieve these goals, resulting in an uneven split for some factors.

All participants provided informed consent and were paid by Qualtrics for completing a survey. All studies were approved by Temple University's Institutional Review Board.

All regression analyses across studies were conducted using iteratively reweighted least-squares robust regression models with Huber weights to compute the coefficient of single factor and interaction effects of our interests. The robust regression approach deviated from our pre-registered analysis plan of using ordinary linear squares (Study 1) and two-way ANOVAs (Study 2 & 3). This deviation was motivated by several factors. First, our key financial exploitation measure (OAFEM) was zero-inflated with extreme outliers, and various tests commonly violated the normality and homoscedasticity assumptions required for ANOVA. Moreover, discretizing a continuous variable into medians would have imposed relatively arbitrary cut-offs, hindering cross-study comparability and complicating interpretation. Robust regression allowed us to downweight the influence of outliers while preserving the continuous nature of our measures, thereby providing a better fit to the majority of the data.

We included age, gender, race, and education as control variables in all studies when they were not among our primary factors of interest in an analysis (e.g., when an analysis involved socioeconomic status, education was not included as a control variable since education was a component of our socioeconomic status composite score). Although this approach deviates from our pre-registration, where we only planned to control for age, we believe it increases rigor and facilitates comparisons with other studies examining financial exploitation[25,37].

For multiple-comparison correction, we adopted the approach in Derringer[38]. This is another deviation from our pre-registration of using Bonferroni correction. We made the decision to accommodate the non-independence among our measures and tests. The approach in Derringer[38] takes into account such non-independence when adjusting for multiple comparisons, offering a more reasonable balance between false positive vs false negative findings compared to the Bonferroni correction, which would be too stringent for our studies, especially given our studies' exploratory nature.

All analyses were performed in R Statistical Software (v 4.3.0, R Core Team[39]). Robust regressions were performed using 'rlm' and its default settings of the MASS R package (v 7.3.60; Venables & Ripley[40]). We adopted a robust $F$-test (i.e., a Wald test for multiple coefficients,) to compute the $p$ value of a correlation coefficient in robust regression using 'f.robftest' and its default settings of the sfsmisc R package (version 1.1-16; Maechler[41]).

Our analyses employed robust regression and Welch's ANOVA, which relax key assumptions of traditional parametric methods. Robust regression does not require formal testing of normality or homoscedasticity, as it downweights outliers and provides reliable estimates when residuals are non-normal or heteroscedastic[42]. This approach was particularly suitable for our zero-inflated dependent variables (OAFEM and FEVS), where classical assumptions were likely violated due to extreme skewness and excess zeros (Delacre et al.[43]). Similarly, Welch's ANOVA does not assume equal variances across groups, making it robust to heteroscedasticity[43]. Formal tests for normality or homogeneity of variances were therefore unnecessary for these methods.

### Deviations from Pre-registrations

There were several deviations from the preregistrations across the studies, including the exclusion of certain pre-registered hypotheses, changes to planned analyses, adjustments to multiple-comparison correction methods, and the inclusion of unregistered interaction analyses. These deviations were made to address methodological challenges, maintain consistency across studies, and ensure the robustness of our findings. A detailed summary of deviations, along with their corresponding reasons, is provided in Table 5. We also provided mappings of the reported results to the pre-registered hypotheses and corresponding results using the originally planned analyses for each study in Tables S11–S22 in the supplementary materials.

### Study 1 - Risk factors in an older adult sample from Philadelphia metropolitan area

Given the focus on older adults in previous financial exploitation research[31], we first focused on risk factors of financial exploitation within older adults from our local community in Philadelphia, PA, USA. Our primary goals were to identify (1) how trust in others and fairness preference across social context in economic activities may be associated with self-reported financial exploitation, and (2) how the association between socioeconomic status and self-reported financial exploitation may be moderated by health-related and psychosocial factors.

**Participants.** A sample of local older adults was recruited using Qualtrics panels. Participants were eligible to enroll in the study if they were 50 years of age or older and currently lived in the Philadelphia Metropolitan Area. The participants were recruited to be evenly split across brackets of household income ($0–50,000; $51,000K–100,000; $101,000 and above), gender (men, women), and education (no degree; college degree; master's degree or above). The final sample included 680 participants. The majority of participants identified as women (57.35%), White (81.18%), and not-Hispanic (96.62%), and ranged in age from 50 to 91 years ($M = 62.48$) (Table 1).

**Materials.** Questionnaires and scales of sociodemographics, financial exploitation, health, economic decisions across social contexts, persuadability, insensitivity to trustworthiness cues, need to belong, perceived social support, and emotion regulation were presented to participants. See Tables for summaries of sociodemographic (Tables 1–3), health-related (Table 4), and psychosocial measures (Table 4). The survey also included several questions and scales that were not intended for the current study and are not reported here. Participants' average total time to complete the survey was 33.32 min.

Sociodemographics. For sociodemographics, we collected participants' education level (Table 2), current monthly expenses, highest annual salary ever earned, annual household income (Table 2), gender, age, race, and ethnicity (Table 1).

Financial exploitation. To measure financial exploitation, we used the 30-item OAFEM (Conrad et al.[1] Table 3). OAFEM measures risk for financial exploitation using 30 items that describe key behaviors related to financial exploitation of different severity (A lesser severity item example: "Have there been unexplained disappearances of your money or possessions?" A major severity item example: "Has someone changed the direct deposit destination so as to benefit themselves?"). Each item asks the participant to indicate whether a potential financial exploitation experience occurred at any point during the past twelve months, including the present. Responses were coded

**Table 1 | Study time, geographical range, and participant demographics**

| | | Study 1 | | Study 2 | | Study 3 | |
|---|---|---|---|---|---|---|---|
| Data collection time | | 2/4–2/29, 2020 | | 4/16–5/21, 2020 | | 7/12–9/10, 2021 | |
| Population | | Philadelphia Metropolitan Area adults of age 50 or above | | Pennsylvania Region adults of age 50 or above | | The United States adults of age 20 or above | |
| Size | | 680 | | 305 | | 844 | |
| Age | | 62.48 (7.89) | | 61.78 (7.95) | | 48.06 (19.57) | |
| | | Count | Percentage | Count | Percentage | Count | Percentage |
| **Gender** | Male | 289 | 42.50% | 120 | 39.34% | 417 | 49.41% |
| | Female | 390 | 57.35% | 185 | 60.66% | 427 | 50.59% |
| | Non-binary | 1 | 0.15% | 0 | 0.00% | 0 | 0.00% |
| **Race** | American Indian or Alaska Native | 6 | 0.88% | 2 | 0.66% | 38 | 4.50% |
| | Asian | 9 | 1.32% | 4 | 1.31% | 74 | 8.77% |
| | Native Hawaiian or Pacific Islandar | 1 | 0.15% | 0 | 0.00% | 5 | 0.59% |
| | White | 552 | 81.18% | 259 | 84.92% | 449 | 53.20% |
| | More than one race | 19 | 2.79% | 10 | 3.28% | 79 | 9.36% |
| | Black or African American | 93 | 13.68% | 30 | 9.84% | 199 | 23.58% |
| Ethnicity | Hispanic or Latino | 23 | 3.38% | 13 | 4.26% | 78 | 9.24% |
| | Not Hispanic or Latino | 657 | 96.62% | 292 | 95.74% | 766 | 90.76% |

using a three-point scale (0 = "No", 1 = "Suspected", 2 = "Yes"). The final OAFEM score is a severity-weighted sum across all 30 items (Entitlement Expectation = 1; Lesser Theft & Scams = 2; Major Theft & Scams = 3). The weighted sum of rated items is used as the total score. Total scores range from 0 to 124 and higher scores reflect higher risk for financial exploitation. OAFEM was originally developed as a self-report measure conducted by an interviewer. In our online study we used OAFEM as a pure participant-driven self-report measure and modified the question prompt accordingly (i.e., "Please select an answer for each question. All questions refer to the past 12 months, including the present."). In addition, due to time constraints and the self-report nature of the survey the subject in each question was referred to in general terms (i.e., "someone").

Trust in others and fairness preference. To assess participants' trust in others and fairness preference in economic activities, we used hypothetical questions from the Trust Game[16] and the Ultimatum Game[15], respectively. We included six items from the Ultimatum Game. Each item asks if the participant would accept or reject an offer from a stranger who shares a proportion of a certain amount of money with the participant (e.g., "Consider the following scenario. Let's imagine Jordan is given $35, but Jordan has to share some of the $35 with you. Jordan offers you $7. If you accept Jordan's offer, you get $7 and Jordan gets $28. If you reject Jordan's offer, neither of you get any money. Accept or Reject?"). The offer in each item varies across the task in terms of fairness (e.g., an offer of 48%–50% split of the money to the participant was coded as fair, whereas offers of less than 25% to the participant were coded as unfair). There were four unfair items and two fair items. We also included four items from the Trust game. Each item asks how much money a participant would invest for a prospective higher return with a person who could choose to either split the multiplied amount evenly (reciprocate) or retain it for themselves (defect) (e.g., "Consider the following scenario. You are given $23. You can keep all of this money or share some of it with Charlie, a stranger you haven't met. Whatever you share with Charlie will be tripled. Importantly, Charlie can keep all of the tripled sum of money to himself or share the tripled sum of money evenly with you. How much would you like to share with Charlie?"). The person was either a hypothetical friend in two of the items or a hypothetical stranger in the other two items (adopted from Fareri et al.[44]). Participants were asked to indicate how much in dollars they would share with the other person (e.g., how much out of $28. The maximum amount varied

between $22, $23, $27, and $28 across four items). We used the amount of money shared with others as an indicator of participants' trust in the other person.

Health-related indicators. To assess health, we adopted the physical health and mental health subscales from PROMIS global items[45] as measures of physical health and mental health. We also adopted the Everyday Cognition Scale as our measure of everyday cognitive functioning.

*Physical and Mental Health Summary Scores*[46]. This scale assesses general perceptions of one's health. The scale consists of four global physical health items on overall physical health ("In general, how would you rate your physical health?"), physical function ("To what extent are you able to carry out your everyday physical activities such as walking, climbing stairs, carrying groceries, or moving a chair?"), pain ("In the past 7 days, how would you rate your pain on average? 0 means 'no pain', 10 means 'worst pain imaginable'"), and fatigue ("In the past 7 days, how would you rate your fatigue on average?"), and four global mental health items on quality of life ("In general, how would you rate your quality of life?"), mental health ("In general, how would you rate your mental health, including your mood and your ability to think?"), satisfaction with social activities ("In general, how would you rate your satisfaction with social activities and relationships?"), and emotional problems ("How often have you been bothered by emotional problems?"). Participants provided their answers using a 5-point scale in which higher score indicates better health. All answered items are summed, with total scores ranging from 4 to 20 and higher scores reflecting better health.

*Everyday Cognition Scale (Ecog)*[47]. This scale is a 12-item global measure of perceived decline of everyday cognitive abilities. The items ask about participants' performance in different domains of everyday function by asking them to compare how they function now compared to 10 years ago (e.g., "Remembering where you have placed objects."). Participants can indicate their performance by choosing from a four-point scale (1 = better or no change; 2 = a little worse sometimes; 3 = a little worse all the time; 4 = much worse; there is also a "don't know" option which was coded as missing data). Ecog was initially developed as an informant-rated tool to measure cognitive decline. Subsequent studies have used it as a self-report measure of subjective cognitive decline and have demonstrated equivalent if not improved performance in predicting mild cognitive decline compared to the informant version (Farias et al.[48,49]; see Rabin et al.[50] for a review). Thus, given the self-reported nature of our survey study, we used the 12-item self-report version of the Ecog as a measure of cognitive decline (e.g., "Please

**Table 2 | Participant education level and personal and household economic information**

| | | Study 1 | | | Study 2 | | | Study 3 | | |
|---|---|---|---|---|---|---|---|---|---|---|
| | | Count | Percentage | Code for analyses | Count | Percentage | Code for analyses | Count | Percentage | Code for analyses |
| Education level | Less than high school | 1 | 0.15% | 1 | 1 | 0.3% | 1 (< = 9 year) | 20 | 2.37% | 1 |
| | Some high school | 2 | 0.29% | 2 | 8 | 2.6% | 2 (9–11 years) | | | |
| | High school diploma/GED | 114 | 16.76% | 3 | 71 | 23.3% | 3 (12 years) | 406 | 48.10% | 2 |
| | Some college (technical school, college, university | 143 | 21.03% | 4 | 22 | 7.2% | 4 (13 years) | | | |
| | Associate's degree or other professional certification (e.g., medical technician) | 62 | 9.12% | 5 | 63 | 20.7% | 5 (14–15 years) | | | |
| | Bachelor's degree | 198 | 29.12% | 6 | 53 | 17.4% | 6 (16 years) | 260 | 30.81% | 3 |
| | Advanced degree | 160 | 23.53% | 7 | 87 | 28.5% | 7 (> = 17 years) | 158 | 18.72% | 4 |
| Total monthly expenses ($) | <= 1000 | 67 | 9.9% | 1 | | | | | | |
| | 1001–2500 | 238 | 35.0% | 2 | | | | | | |
| | 2501–4000 | 183 | 26.9% | 3 | | | | | | |
| | 4001–5500 | 67 | 9.9% | 4 | | | | | | |
| | 5501–7000 | 58 | 8.5% | 5 | | | | | | |
| | 7001–8500 | 13 | 1.9% | 6 | | | | | | |
| | >8500 | 29 | 4.3% | 7 | | | | | | |
| | Not sure | 0 | 0.0% | 0 | | | | | | |
| Total annual household income ($) | <12500 | 31 | 4.6% | 1 | | | | 89 | 10.5% | 1 |
| | 12500– 18750 | 28 | 4.1% | 2 | | | | 52 | 6.2% | 2 |
| | 18751– 28120 | 54 | 7.9% | 3 | | | | 102 | 12.1% | 3 |
| | 28121– 50000 | 133 | 19.6% | 4 | | | | 168 | 19.9% | 4 |
| | 50001– 63280 | 60 | 8.8% | 5 | | | | 110 | 13.0% | 5 |
| | 63281– 99999 | 149 | 21.9% | 6 | | | | 148 | 17.5% | 6 |
| | 100000– 142380 | 108 | 15.9% | 7 | | | | 108 | 12.8% | 7 |
| | >142380 | 117 | 17.2% | 8 | | | | 67 | 7.9% | 8 |
| Highest annual salary ($) | <12500 | 17 | 2.5% | 1 | 40 | 13.1% | 1 (< 13k) | 82 | 9.7% | 1 |
| | 12500– 18750 | 26 | 3.8% | 2 | 16 | 5.2% | 2 (13–19k) | 62 | 7.3% | 2 |
| | 18751– 28120 | 55 | 8.1% | 3 | 41 | 13.4% | 3 (19–29k) | 102 | 12.1% | 3 |
| | 28121– 50000 | 98 | 14.4% | 4 | 34 | 11.1% | 4 (29–42k) | 147 | 17.4% | 4 |
| | 50001– 63280 | 143 | 21.0% | 5 | 53 | 17.4% | 5 (42–63k) | 154 | 18.2% | 5 |
| | 63281– 99999 | 145 | 21.3% | 6 | 49 | 16.1% | 6 (63–95k) | 131 | 15.5% | 6 |
| | 100000– 142380 | 115 | 16.9% | 7 | 37 | 12.1% | 7 (95–140k) | 104 | 12.3% | 7 |
| | >142380 | 81 | 11.9% | 8 | 35 | 11.5% | 8 (> 140k) | 62 | 7.3% | 8 |

**Table 3 | Financial exploitation risk measures**

| Measure | Literature | Range | Study 1 | Study 2 | Study 3 |
|---|---|---|---|---|---|
| Older Adults Financial Exploitation Measure (OAFEM) | Conrad et al.[1] | 0–124 | ◯ | ◯ | ◯ |
| Short Form of Financial Exploitation Vulnerability Scale (FEVS-SF) | Campbell & Lichtenberg[14] | 0–18 | | | ◯ |

"◯" indicates a study used the measure.

rate your ability to perform certain everyday tasks NOW, as compared to 10 years ago"). The sum of all ratings is divided by the number of items completed, with total scores ranging from 1 to 4 and higher scores reflecting greater decline.

Psychosocial factors. We employed several self-report measures to examine different psychosocial factors of everyday cognitive functioning, social experience and social need, emotion regulation, and gullibility.

*Need to belong scale*[51]. This 10-item scale measures participants' desire for acceptance and belonging by asking how much they agree with specific description of their related behaviors (e.g., "I try hard not to do things that will make other people avoid or reject me.") Participants indicate how much they agree with the description using a five-point scale (1 = strongly disagree; 2 = moderately disagree; 3 = neither agree nor disagree; 4 = moderately agree; 5 = strongly agree). The sum of all ratings is used as total scores ranging from 5 to 50 and higher scores reflecting higher need to belong.

*Perceived social support*[52]. The 12-item scale measures participants' subjective experience of social support by asking how much they agree with specific descriptions of their related experience (e.g, "My family really tries to help me."). Participants indicate how much they agree with the description using a seven-point scale (1 = very strongly disagree; 2 = strongly disagree; 3 = mildly disagree; 4 = neutral; 5 = mildly agree; 6 = strongly agree; 7 = very strongly agree). The sum of all ratings is divided by the number of items completed, with total scores ranging from 1 to 7 and higher scores reflecting higher perceived social support.

*Persuadability and insensitivity to untrustworthiness cues subscales*[53]. We used these two subscales from the two-factor Gullibility Scale to measure participants' propensity to be persuaded and insensitivity to the presence of untrustworthiness cues. Each subscale has six items asking how much the participant agree with specific descriptions of their related behaviors (e.g, "Your family thinks you are an easy target for scammers" from the persuadability subscale; "You are pretty poor at working out if someone is tricking you" from the insensitivity to trustworthiness cues subscale). Participants indicate how much they agree with the description using a seven-point scale (1 = strongly disagree; 2 = disagree; 3 = somewhat disagree; 4 = neither agree nor disagree; 5 = somewhat agree; 6 = agree; 7 = strongly agree). The sum of all ratings is used as total scores of each subscale ranging from 6 to 42 and higher scores reflecting higher persuadability or insensitivity to trustworthiness cues.

*Emotional regulation of others and self*[54]. We used this measure to evaluate emotion regulation of improving or worsening the feelings of self (i.e., intrinsic affect) of others (extrinsic affect). For each type of emotion regulation (i.e., intrinsic affect improving, e.g., "I thought about my positive characteristics to make myself feel better"; intrinsic affect worsening, e.g., "I looked for problems in my current situation to make myself feel worse"; extrinsic affect improving, e.g., "I gave someone helpful advice to try to improve how they felt"; and extrinsic affect worsening, e.g., "I told someone about their shortcomings to try to make them feel worse"), participants are asked to report how much they had used specific strategies try to change their own feelings (intrinsic items) or someone else's feelings (extrinsic items) (e.g., a extrinsic affect improving item: "I gave someone helpful advice to try to improve how they felt"). Participants answer by choosing from a five-point scale of different amounts of effort in using a strategy (1 = not at all; 2 = just a little; 3 = moderate amount; 4 = quite a bit; 5 = a great deal). The sum of all ratings is divided by the number of items completed, with total scores ranging from 1 to 5 and higher scores reflecting greater tendency to regulate the emotion of oneself or others.

**Statistical analyses**. One focus of Study 1 was to investigate how trust in others and fairness preference during economic games may be associated with self-reported financial exploitation. We measured participants' evaluation of fairness from the Ultimatum Game trials using the difference in the proportion of rejection of unfair and fair monetary offers. We measured trust in friends and in strangers using the average proportion of money shared with each partner type during the Trust Game. We also used the sum of the two proportions as a measure of overall trust in other people and the difference between the two proportions as a measure of a participant's differentiation of trust in strangers versus friends. We hypothesized that: (1) trust in others and financial exploitation would be positively correlated; and (2) evaluation of fair and unfair monetary offers would be associated with financial exploitation. Table S1 contains all the pre-registered (https://osf.io/fcrus, February 17th, 2020) analyses we conducted to test for effects of trust in others and fairness preference during economic games.

The second focus of Study 1 was to examine how the association between socioeconomic status and self-reported financial exploitation may be moderated by health-related and psychosocial factors, such as emotion regulation of self and others. To measure socioeconomic status, we first categorized and ordinally coded one's highest lifetime annual salary, current monthly expenses, education level, and current annual household income (see Table 2 for categories and coding of each socioeconomic status item).

We then used the sum of all coded items as a composite index for socioeconomic status. We hypothesized that people with lower socioeconomic status will be more vulnerable to risk factors of poor health, larger cognitive decline, lower perceived social support, higher need to belong, and poorer emotion regulation. For example, we predicted that cognitive decline would be positively associated with self-reported exploitation, and that this association would be stronger among individuals from lower versus higher SES. Table S2 presents all the hypothesized interactions between socioeconomic status and health-related and psychosocial factors. The scores for each health-related and psychosocial factor are calculated according to their original published description (see Table 4 for citation details).

## Study 2 - Risk factors in an older adult sample from Pennsylvania region

In Study 2, we sought to replicate our findings from Study 1 in a more diversified population within the Pennsylvania region, USA. Our primary goals were the same as in Study 1: To evaluate (1) how trust in others and fairness preference during economic games may be associated with self-reported financial exploitation, and (2) how the association between socioeconomic status and self-reported financial exploitation may be moderated by health-related and psychosocial factors.

**Participants**. Similar to Study 1, a sample of older adults from the Pennsylvania region was recruited using Qualtrics panels. Participants were eligible to enroll in the study if they were 50 years of age or older and currently lived in the Pennsylvania region. The participants were drawn from various urban and rural areas across the state that encompass varying levels of household incomes and were evenly split based on gender (men and women). Qualtrics distributed the survey to participants based on their eligibility and the targeted income, geographic area, and gender brackets. The final sample included 305 participants. The majority of participants identified themselves as women (60.66%), white (84.92%) and not-Hispanic (95.74%), and ranged in age from 50 to 94 (*M* = 61.78) (Table 1).

**Materials**. We retained most measures from Study 1 (Tables 1–4). However, we do note a few key differences to the trust in others and fairness preference measures and our collection of socioeconomic status-related metrics that were made in order to minimize survey length and remove superfluous items to increase participant retention. In contrast to Study 1 where we used six items for the Ultimatum Game, for Study 2 we only assessed participants' evaluation of one unfair offer from a hypothetical stranger. Similarly, instead of four items for the Trust Games as in Study 1, in Study 2 we kept only one item of sharing money with a hypothetical stranger. Also, in Study 2 we used a composite score of participant's education and maximum individual annual income as our measure of socioeconomic status instead of ZIP code based household income as we planned in our pre-registration. Although this measurement of socioeconomic status deviates from our pre-registration, we believe it facilitates comparisons across studies when examining effects of socioeconomic status. Sociodemographics (Tables 1 and 2), health-related and psychosocial measures (Table 4) were retained in their original form from Study 1. The survey also included several questions and scales that are for other studies and are not reported here. Participants' average total time to complete the survey is 32.98 min.

**Statistical analyses**. Tables S3 and S4 contain summaries of the pre-registered analyses (https://osf.io/v357h, May the 11th, 2020) we conducted for Study 2. For single-factor effects of our trust in others and fairness preference measures, we hypothesized that: (1) trust in others (as measured by proportion of money shared with the hypothetical stranger) and financial exploitation are positively correlated; and (2) fairness preference (as measured by accepting or rejecting the unfair monetary offer) is associated with financial exploitation. For interactions with socioeconomic status, we hypothesized that socioeconomic status would

**Table 4 | Health-related and psychosocial measures**

| Measure | Literature | Range | Study 1 | Study 2 | Study 3 |
|---|---|---|---|---|---|
| Measurement of everyday cognition (ECog) | Farias et al., 2011 | 1–7 | ◯ | ◯ | ◯ |
| Patient—Reported outcomes measurement information system (PROMIS) physical health items | Hays et al.[46] | 5–20 | ◯ | ◯ | ◯ |
| Patient—Reported outcomes measurement information system (PROMIS) mental health items | Hays et al.[46] | 5–20 | ◯ | | ◯ |
| Trust in others (Dictator Games) | Berg et al.[16] | 0–1 | ◯ | ◯ (trust in strangers only, one question) | ◯ (one question for friend and one question for stranger) |
| Perceived fairness (Ultimatum Games) | Güth et al.[15] | 0–1 | ◯ | ◯ (fair offers only) | ◯ (fair offer only, one question) |
| Multidimensional scale of perceived social support | Zimet et al.[52] | 1–7 | ◯ | ◯ | ◯ |
| Need to belong scale | Leary et al.[51] | 1–5 | ◯ | | ◯ |
| Persuadability | Teunisse et al.[53] | 1–7 | ◯ | ◯ | ◯ |
| Insensitivity to trustworthiness cues | Teunisse et al.[53] | 1–7 | ◯ | ◯ | ◯ |
| Intrinsic affect-improving | Niven et al.[54] | 1–5 | ◯ | ◯ | |
| Intrinsic affect-worsening | Niven et al.[54] | 1–5 | ◯ | ◯ | |
| Extrinsic affect-improving | Niven et al.[54] | 1–5 | ◯ | ◯ | |
| Extrinsic affect worsening | Niven et al.[54] | 1–5 | ◯ | ◯ | |
| Cognitive Reflection Test (CRT) | Frederick[55]; Thomson & Oppenheimer[56] | 0–7 | | | ◯ |

"◯" indicates a study used the measure.

have statistically significant interactions with health, cognitive decline, perceived social support, and emotion regulation.

### Study 3 - Risk factors in a national sample of both younger and older adults

Building off findings of interactions between socioeconomic status and several psychosocial factors from Study 1 and Study 2 that included only older people from Philadelphia metropolitan area and Pennsylvania region and the majority of participants were White, one open question relates to how demographic factors such as age, gender and race, and health-related factors would interact with psychosocial factors in a more diverse sample? Therefore in Study 3 we expanded our sample's age range, racial diversity, and geographical range, and examined how sociodemographic and health-related factors may moderate the association between psychosocial factors and financial exploitation.

**Participants.** A national sample of adults was recruited using Qualtrics panels. Participants were eligible to enroll in the study if they were 20 years of age or older and currently lived in the United States. The participants were drawn to be evenly split across brackets of household income ($0–50,000; $51,000K–100,000; $101,000 and above), gender (men and women), race, and education (no degree; college degree; master's degree or above). The final sample included 933 participants. The majority of participants identified themselves as men (50.05%), White (53.91%), and not-Hispanic (91.10%), and ranged in age from 20 to 94 ($M = 46.83$) (Table 1).

**Materials.** In Study 3 we again retained most scales from the previous two studies (Table 1). One key change was made to the trust in others and fairness preference measures during economic games due to survey length changes. Similar to Study 2, we used one item of unfair offer from a hypothetical stranger for the Ultimatum Game. Different from Study 2 where we used one item of sharing money with a hypothetical stranger from the Trust Game, in Study 3 we used two items of monetary sharing, one with a hypothetical friend and one with a hypothetical stranger.

We also added two new measures. First, in order to capture a second assessment of risk for financial exploitation, we also administered The FEVS-SF (Campbell & Lichtenberg[14], Table 3). FEVS-SF is a nine-item scale

that evaluates how people make decisions about financial transactions in the real world. Each FEVS-SF item asks about different real-world experiences regarding a participant's personal finances; responses were coded using a three-point scale wherein response options differed according to the question (e.g., "How confident are you in making big financial decisions?", 0 = confident; 1 = unsure; 2 = not confident; "How often do you feel anxious about your financial decisions and/or transactions?", 0 = never or rarely; 1 = sometimes; 2 = often). The final FEVS-SF score is a sum across all nine items ranging from 0 to 18 and higher scores indicating higher financial exploitation vulnerability. Second, we added an additional measure of cognitive function, the Cognitive Reflection Test (Frederick[55]; Thomson & Oppenheimer[56]) which measures the ability or disposition to resist reporting an intuitive response that occurs to one first yet may be wrong (e.g., "How many cubic feet of dirt are there in a hole that is 3 feet deep × 3 feet wide × 3 feet long? Please answer using a number. For example, please type "10" instead of "ten." The correct answer is 0 (zero). An example intuitive answer is 27).

**Statistical analyses.** In Study 3 our focus was twofold. First, we studied which psychosocial variables were associated with financial exploitation. Second, we investigated how the associations between psychosocial variables and financial exploitation might be moderated by sociodemographic and health-related variables. Tables 1–5 contain all the psychosocial, sociodemographic, and health-related variables we planned to use for Study 3 analyses (pre-registration: https://aspredicted.org/blind.php?x=TPQ_UDJ, April 29th, 2021). For socioeconomic status, similar to Study 1 we used the coded sum of one's highest lifetime annual salary, education level, and annual household income as our composite index of socioeconomic status (Table 2).

We predicted (1) psychosocial variables tied to trust, fairness, social support, cognitive reflection, and need to belong will be associated with risk for financial exploitation, and (2) these relationships will be moderated by sociodemographic factors such as age, socioeconomic status, and race/ethnicity and also health-related variables of cognitive decline, physical health, and mental health. To test the hypotheses, we used the same robust regression approach with age as a covariate and multiple comparison correction as in Study 1 and Study 2. We note that we tested our hypotheses using both OAFEM and FEVS-SF, respectively as two distinct measures of financial exploitation with non-overlapping items. We originally expected a

## Table 5 | Deviations from Pre-registrations

| Study | Area | Description | Reason |
|---|---|---|---|
| 1 | Correction for multiple comparison | We used the approach in Derringer[38] instead of Bonferroni correction. | We used the approach in Derringer[38] to account for the non-independence among our tests. This approach offers a more balanced trade-off between false positives and false negatives. |
| 1 | Hypothesis testing and reporting | Hypotheses related to impulsivity were not reported in the manuscript. | Data for testing the hypotheses were only available in Study 1 but not in Studies 2 and 3 due to funding and survey length constraints. The lack of data for Studies 2 and 3 limited our ability to compare results across studies and draw convincing conclusions. Consequently, we chose not to report this hypothesis in the manuscript. |
| 1 & 2 | | Hypotheses related to family history of Alzheimer's Disease (AD) or Alzheimer's related diseases (ADRD) were not tested or reported in the manuscript. | Evidence from the literature revealed significant challenges in obtaining reliable diagnosis of AD or dementia (e.g., Amjad et al., 2018). We were not initially aware of the difficulty in accurately measuring AD and ADRD, which prevented us from making convincing conclusions. As a result, we decided to exclude this hypothesis from testing. |
| 1 & 2 | | Interaction analyses examining the relationship between emotion regulation and socioeconomic status were not specified in the preregistration. | These interaction hypotheses were unintentionally omitted from the preregistration due to an oversight, despite the corresponding main effects for emotion regulation and socioeconomic status being explicitly specified. Including these analyses aligns with our research objectives. |
| 2 | | Hypotheses that were planned for investigating interactions that do not involve self-reported SES (i.e., Hypotheses d, f, g, h, i) were not reported. | These hypotheses were excluded because they were not aligned with the primary purpose of Study 2. The purpose of Study 2 is to replicate our findings from Study 1. Including these hypotheses would have deviated from the study's intended focus and objectives. |
| 3 | | We used persuadability and insensitivity to trust cues as independent variables rather than as proxies for fraud (the dependent variable), as initially planned. | In Studies 2 and 3 analyses, persuadability and insensitivity to trust cues were incorporated as independent variables rather than proxies for fraud. To maintain methodological consistency across all three studies and facilitate direct comparison of results to draw convincing conclusions, we applied this approach to Study 1 as well. |
| 1, 2 & 3 | Analysis | Instead of ordinary linear squares (Study 1) and two-way ANOVAs that would categorize continuous variables using zeros and/or medians (Study 2 & 3), we used robust regression for all primary analyses. | This deviation was motivated by several factors. First, our key financial exploitation measure (OAFEM) was zero-inflated with extreme outliers, and various tests commonly violated the normality and homoscedasticity assumptions required for ANOVA. Moreover, discretizing a continuous variable into medians would have imposed relatively arbitrary cut-offs, hindering cross-study comparability and complicating interpretation. Robust regression allowed us to downweight the influence of outliers while preserving the continuous nature of our measures, thereby providing a better fit to the majority of the data. |
| 3 | Participant exclusion | Instead of planned exclusion of participants whose OAFEM and FEVS-SF risk categorizations (none, low, high) differed, we excluded participants whose z-transformed OAFEM and FEVS-SF scores show an absolute difference of two or above. | After planned exclusion, only 420 out of 933 participants remained. We deviated from the planned exclusion to balance between retaining enough power and having financial exploitation measured more reliably. After the deviated exclusion, 844 out of 933 participants remained. |

strong correlation between OAFEM and FEVS-SF and planned to include only participants who scored similarly on both measures. We respectively binned OAFEM and FEVS-SF into three levels: no risk of fraud (score = 0), low risk of fraud (score < median for non-zero scores), high risk of fraud (score >= median for non-0 scores). Among 933 participants, only 420 participants ($M = 45.73$, $SD = 19.46$) scored similarly (i.e., fell into the same bin) on both measures, and a follow-up correlation analysis using the full sample of 933 participants revealed only a moderate correlation between OAFEM and FEVS-SF ($r = 0.33$, $p < 0.001$).

To balance between retaining enough power and having financial exploitation measured more reliably, we decided to exclude participants whose z-transformed OAFEM and FEVS-SF scores show an absolute difference of two or above. After exclusions, our final sample size was 844 ($M = 48.06$, $SD = 19.57$). We expected similar results using either OAFEM or FEVS-SF as independent variables, respectively for this sample of 844 participants after filtering. However, the results were not consistent, especially the interaction results. Therefore, we report results from the 844 participants using OAFEM and FEVS-SF as our measures of financial exploitation separately below.

**Post-hoc analyses of financial exploitation measures**

One pattern that unexpectedly stood out across three studies is the higher mean of OAFEM in Study 3 (5.4 in Study 1, 4.3 in Study 2, and 14.3 in Study 3). We explored the difference in Study 3 OAFEM scores in two ways. (1) To determine whether higher scores within Study 3 were elevated due to differences in one or the other age sample, we conducted a Welch two-sample t test comparing scores in younger and older adults. (2) We conducted a Welch one-way ANOVA of OAFEM scores among older adults across the three studies to determine if there was a study effect.

Another pattern that unexpectedly stood out from Study 3 is the additional risk measure of FEVS score we included in Study 3 were relatively higher than OAFEM score (FEVS with a mean of 5 on a 0–18 scale vs OAFEM with a mean of 14.25 on a 0–115 scale) despite that both scores are developed to measure financial exploitation risk. To explore the difference, we normalized FEVS and OAFEM into percentages of each measure's maximum possible range using each measure's maximum possible score. We then conducted a repeated measures regression to examine potential effects of measure type (FEVS vs OAFEM) and age (younger adults of age

below 50 vs older adults of age 50 or above). We also conducted item-wise analyses to explore item-level patterns of age.

### Reporting summary
Further information on research design is available in the Nature Portfolio Reporting Summary linked to this article.

## Results
### Study 1 - Risk factors in an older adult sample from Philadelphia metropolitan area
To test the association between trust in others and fairness preference during economic games and risk for financial exploitation, we constructed models regressing OAFEM onto a social-economic measure with age included as a covariate. Table S1 contains a summary of estimates of regression coefficients as well as descriptive statistics for regressors. We did not find significant associations between OAFEM and trust in other people or evaluations of fairness in others (all $p$-values > 0.15).

To test the interaction between socioeconomic status and other health-related and psychosocial measures, for each potential interaction pair (e.g., socioeconomic status * physical health), we built a model that regressed OAFEM onto socioeconomic status, a health-related or psychosocial measure of interest, and the interaction between the two variables, with age included as a covariate. Table S2 presents a summary of estimates of regression coefficients of interaction terms as well as descriptive statistics for regressors. Results revealed a significant interaction between socioeconomic status and the intrinsic affect-worsening component of our emotion regulation measures ($b = -0.26$, $f^2 = 0.008$, 95%CI = [0,0.095], $p < 0.001$). In other words, greater risk for financial exploitation is positively associated with a tendency to deliberately worsen one's own feelings, and this association is stronger for individuals of lower socioeconomic status than higher socioeconomic status (Fig. 1). No other interactions reached statistical significance.

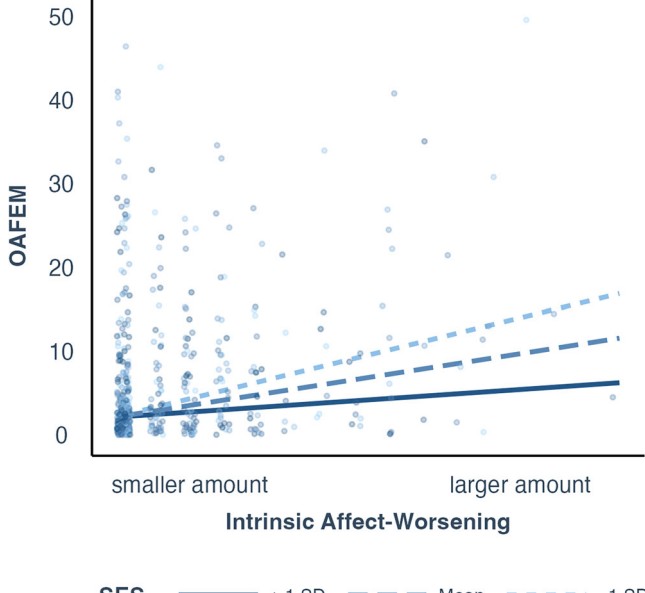

**Fig. 1 | Study 1 SES interactions.** Scatterplot ($n = 680$ participants) with regression lines showing associations between OAFEM (Older Adult Financial Exploitation Measure) and the interaction between emotion regulation and SES (socioeconomic status). Regression lines represent interaction between intrinsic affect-worsening and SES with age, race, gender are controlled. Different lines represent the relationship between the variables at different levels of the moderator: the dark solid line shows +1 SD (standard deviation) above the mean, the dashed line represents the mean, and the light dashed line indicates −1 SD below the mean of the moderator. To facilitate visualization, data points are jittered horizontally, and 4 data points (out of 680 total) with OAFEM larger than 50 are not shown.

Following the non-significant interactions between socioeconomic status and the remaining health-related and psychosocial variables, we explored their association with OAFEM (see Table S1 for a summary of estimates). Results from these follow-up simple main effect analyses revealed that, for health-related factors, higher OAFEM was associated with worse physical health ($b = -0.13$, $f^2 = 0.022$, 95%CI = [0.007,0.048], $p < 0.001$), worse mental health ($b = -0.14$, $f^2 = 0.019$, 95%CI = [0.006,0.039], $p < 0.001$), and greater everyday cognitive decline ($b = 0.16$, $f^2 = 0.030$, 95%CI = [0.011,0.071], $p < 0.001$). Among psychosocial factors, higher OAFEM was associated with lower perceived social support ($b = -0.10$, $f^2 = 0.010$, 95%CI = [0.003,0.025], $p < 0.001$) and stronger extrinsic affect-improving ($b = 0.07$, $f^2 = 0.006$, 95%CI = [0.002,0.016], $p < 0.001$), extrinsic affect-worsening ($b = 0.06$, $f^2 = 0.004$, 95%CI = [0,0.018], $p < 0.001$), and intrinsic affect-improving ($b = 0.06$, $f^2 = 0.005$, 95%CI = [0.001,0.014], $p < 0.001$). Higher OAFEM was also associated with higher persuadability ($b = 0.17$, $f^2 = 0.036$, 95%CI = [0.014,0.073], $p < 0.001$) and higher insensitivity to trustworthiness cues ($b = 0.10$, $f^2 = 0.011$, 95%CI = [0.003,0.026], $p < 0.001$). No significant association between OAFEM and socioeconomic status was found.

### Study 2 - Risk factors in an older adult sample from Pennsylvania region
In Study 2, we sought to replicate our findings from Study 1 in a more diversified population within the Pennsylvania region, USA. Our primary goals were the same as in Study 1: To evaluate (1) how trust in others and fairness preference during economic games may be associated with self-reported financial exploitation, and (2) how the association between socioeconomic status and self-reported financial exploitation may be moderated by health-related and psychosocial factors.

We first tested the associations between financial exploitation and trust in others and fairness preference. Associations between trust in others and fairness preference during economic games and financial exploitation are shown in Table S3. Similar to Study 1, we did not find significant associations between OAFEM and proportion of money shared with a hypothetical stranger or rejecting an unfair monetary offer.

We then tested our hypothesized interactions between socioeconomic status and health-related and psychosocial factors. Table S4 presents a summary of estimates of regression coefficients of interaction terms as well as descriptive statistics for regressors. Similar to Study 1, we again found that OAFEM is positively associated with intrinsic affect-worsening, and this positive association is stronger for individuals of lower socioeconomic status than higher socioeconomic status ($b = -0.57$, $f^2 = 0.024$, 95%CI = [0,0.176], $p < 0.001$, Fig. 2A). We also found several other moderating effects of socioeconomic status on the association between extrinsic affect-worsening, everyday cognitive decline, and persuadability (Fig. 2B–D). Compared to individuals with higher socioeconomic status, those with lower socioeconomic status showed stronger positive associations between OAFEM and extrinsic affect-worsening ($b = -0.50$, $f^2 = 0.051$, 95%CI = [0,0.367], $p < 0.001$), everyday cognitive decline ($b = -0.63$, $f^2 = 0.007$, 95%CI = [−0.005,0.099], $p < 0.001$), and persuadability ($b = -0.04$, $f^2 = 0.025$, 95%CI = [0,0.273], $p < 0.001$).

Following the non-significant interactions between socioeconomic status and the rest of health-related and psychosocial variables, we explored their association with OAFEM (see Table S3 for a summary of estimates). Similar to Study 1, higher OAFEM scores were associated with worse physical health ($b = -0.13$, $f^2 = 0.017$, 95%CI = [0.003,0.056], $p < 0.001$) and higher insensitivity to trustworthiness cues ($b = 0.07$, $f^2 = 0.004$, 95%CI = [0,0.017], $p < 0.001$). Also similar to Study 1, no significant association between OAFEM and socioeconomic status was found.

### Study 3 - Risk factors in a national sample of both younger and older adults
Using OAFEM as a measure for financial exploitation, we first examined associations between OAFEM and the psychosocial factors of interest (Table S5). We found higher OAFEM scores were associated with lower

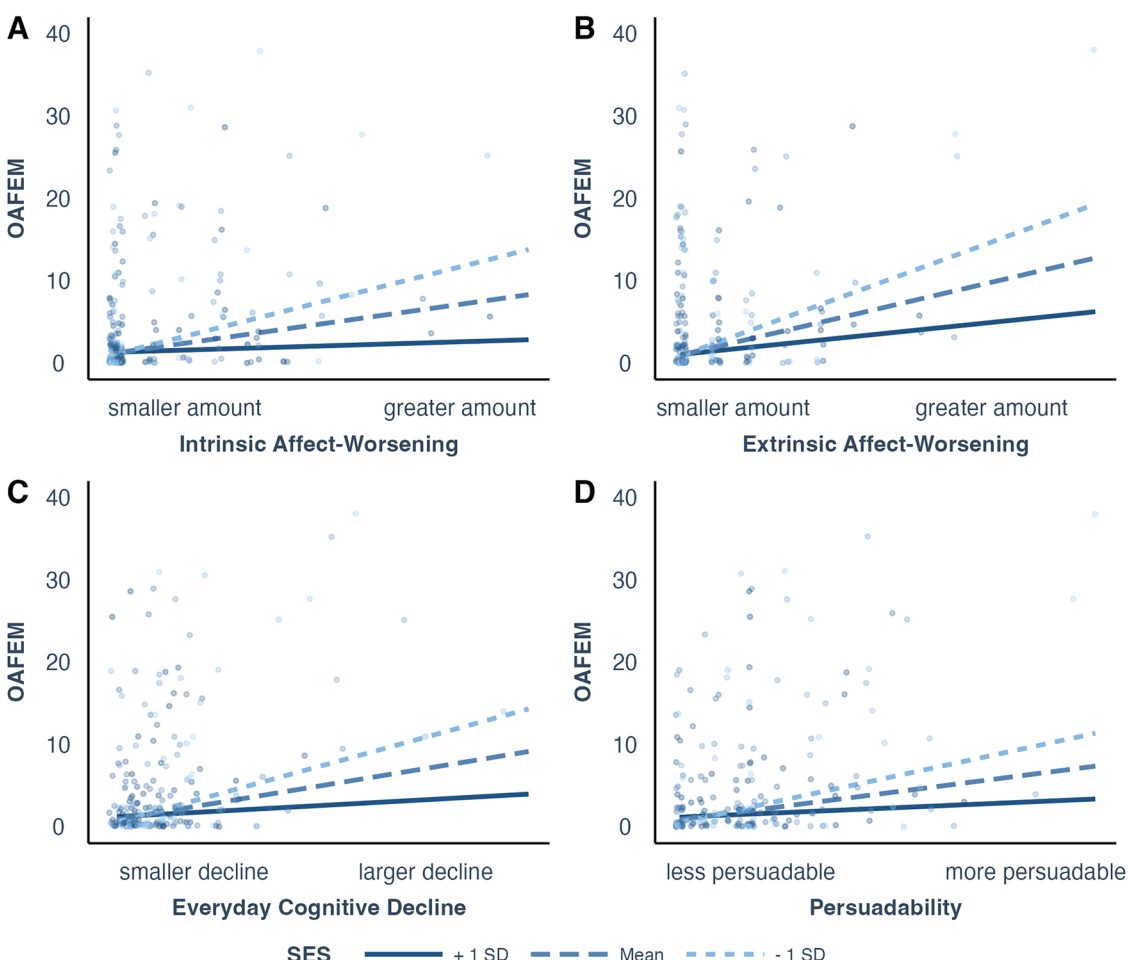

**Fig. 2 | Study 2 SES interactions.** Scatterplots ($n$ = 305 participants) with regression lines showing associations between OAFEM (Older Adult Financial Exploitation Measure) and the interaction between risk factors. Regression lines represent interaction between SES (socioeconomic status) and intrinsic affect-worsening (**A**), extrinsic affect-worsening (**B**), everyday cognitive decline (**C**), and persuadability (**D**) with age, race, and gender controlled. Different lines represent the relationship between the variables at different levels of the moderator: the dark solid line shows +1 SD (standard deviation) above the mean, the dashed line represents the mean, and the light dashed line indicates −1 SD below the mean of the moderator. To facilitate visualization, data points are jittered horizontally and 3 data points (out of 305 total) with OAFEM larger than 40 are not shown.

perceived social support ($b = -0.09$, $f^2 = 0.011$, 95%CI = [0.002,0.026], $p < 0.001$), increased need to belong ($b = 0.08$, $f^2 = 0.010$, 95%CI = [0.002,0.022], $p < 0.001$), increased persuadability ($b = 0.21$, $f^2 = 0.057$, 95% CI = [0.025,0.098], $p < 0.001$), and greater insensitivity to trustworthiness cues ($b = 0.14$, $f^2 = 0.035$, 95%CI = [0.015,0.061], $p < 0.001$). Similar to Study 1 and Study 2, we did not find significant associations between OAFEM and rejecting an unfair monetary offer or between OAFEM with the proportion of money shared with a hypothetical stranger or friend.

Following the significant associations between OAFEM and perceived social support, need to belong, persuadability, and insensitivity to trustworthiness cues, we then tested for potential moderation of sociodemographic and health-related factors on these associations after controlling for age. Table S6 presents all the moderation analyses and results. Compared to younger individuals, older individuals showed weaker positive association between OAFEM and insensitivity to trustworthiness cues ($b = -0.01$, $f^2 = 0.008$, 95%CI = [0,0.023], $p = 0.003$, Fig. 3A) and persuadability ($b = -0.01$, $f^2 = 0.018$, 95%CI = [0.003,0.057], $p = 0.001$, Fig. 3B). We also found the positive association between OAFEM and persuadability is stronger for people with higher SES than people with lower SES ($b = 0.04$, $f^2 = 0.015$, 95%CI = [0,0.063], $p = 0.002$, Fig. 4). We did not find any other significant moderation of sociodemographic and health-related factors.

We then explored the effects of sociodemographic and health-related factors that showed no interactions with psychosocial factors above. We

found that when controlling for age, gender, race, and education, similar to what we found in Study 1 higher OAFEM was associated with worse physical health ($b = -0.10$, $f^2 = 0.015$, 95%CI = [0.005,0.032], $p < 0.001$). We also found, similarly to Study 1 and Study 2, higher OAFEM was associated with more everyday cognitive decline ($b = 0.25$, $f^2 = 0.087$, 95%CI = [0.031,0.146], $p < 0.001$) (Table S5).

We also tested our hypotheses using FEVS-SF as a measure for financial exploitation. Again we first examined associations between FEVS-SF and the psychosocial factors of interest after controlling for age, gender, race, and education (Table S7). Similar to OAFEM, higher FEVS-SF was associated with lower perceived social support ($b = -0.20$, $f^2 = 0.047$, 95% CI = [0.014,0.075], $p < 0.001$), higher need to belong ($b = 0.16$, $f^2 = 0.027$, 95%CI = [0.007,0.052], $p < 0.001$), higher persuadability ($b = 0.28$, $f^2 = 0.092$, 95%CI = [0.043,0.125], $p < 0.001$), and higher insensitivity to trustworthiness cues ($b = 0.23$, $f^2 = 0.058$, 95%CI = [0.021,0.088], $p < 0.001$). Also similar to OAFEM, we did not find significant associations between FEVS-SF and rejecting an unfair monetary offer or between FEVS-SF with the proportion of money shared with a hypothetical stranger or friend.

Following the significant associations between FEVS-SF and perceived social support, need to belong, persuadability, and insensitivity to trustworthiness cues, we then tested for potential moderation of sociodemographic and health-related factors on these significant associations.

**Fig. 3 | Study 3 age interaction, OAFEM.** Scatterplots ($n = 844$ participants) with regression lines showing associations between OAFEM (Older Adult Financial Exploitation Measure) and the interaction between risk factors. Regression lines represent interaction between age and insensitivity to trustworthiness cues (**A**) and age and persuadability (**B**) with race, gender, and education controlled. Different lines represent the relationship between the variables at different levels of the moderator: the dark solid line shows +1 SD (standard deviation) above the mean, the dashed line represents the mean, and the light dashed line indicates −1 SD below the mean of the moderator.

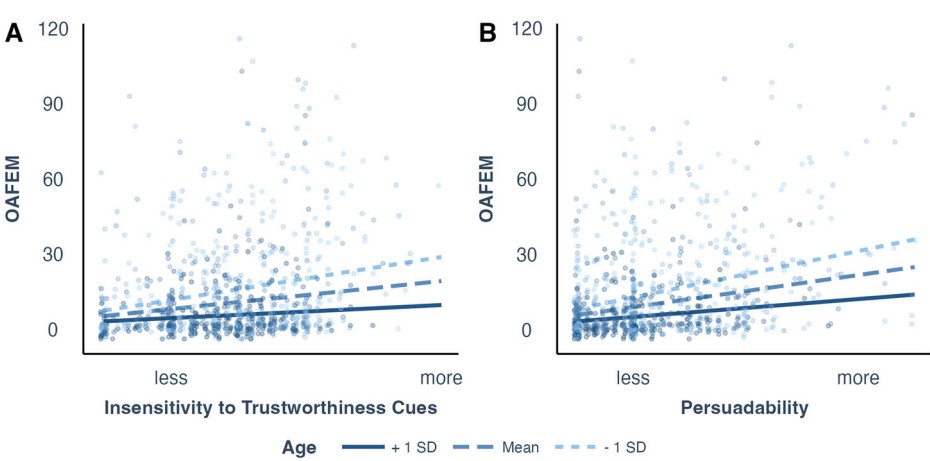

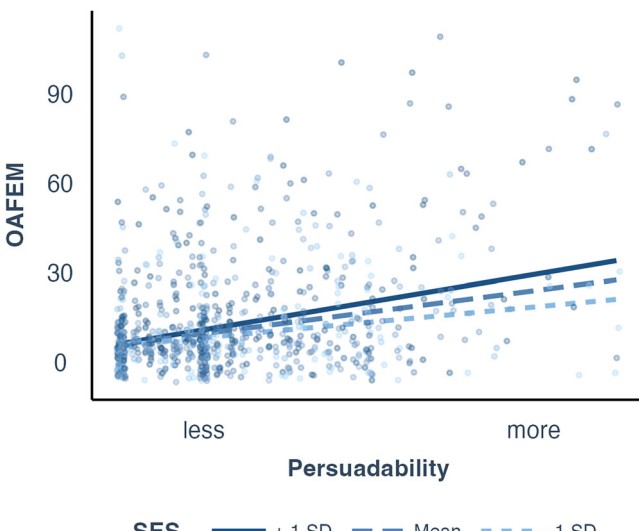

**Fig. 4 | Study 3 SES interaction, OAFEM.** Scatterplots ($n = 844$ participants) with regression lines showing associations between OAFEM (Older Adult Financial Exploitation Measure) and the interaction between risk factors. Regression lines represent interaction between SES (socioeconomic status) and persuadability with age, race, and gender controlled. Different lines represent the relationship between the variables at different levels of the moderator: the dark solid line shows +1 SD (standard deviation) above the mean, the dashed line represents the mean, and the light dashed line indicates −1 SD below the mean of the moderator.

Table S8 presents all the moderation analyses and results. Different from OAFEM, we found the positive associations between FEVS-SF and persuadability and insensitivity to trustworthiness cues were weaker for people reporting higher everyday cognitive decline ($b = -0.12$, $f^2 = 0.016$, 95% CI = [0.002,0.027], $p = 0.001$; $b = -0.19$, $f^2 = 0.019$, 95% CI = [0.003,0.032], $p < 0.001$, Fig. 5). We did not find any other significant moderation of sociodemographic and health-related factors.

We then explored the effects of sociodemographic and health-related factors that showed no interactions with psychosocial factors above (Table S7). We found higher FEVS-SF was associated with younger age ($b = -0.39$, $r^2 = 0.183$, 95% CI = [0.122,0.251], $p < 0.001$). We also found higher FEVS-SF was associated with lower socioeconomic status ($b = -0.20$, $f^2 = 0.050$, 95% CI = [0.018,0.075], $p < 0.001$), lower physical health ($b = -0.35$, $f^2 = 0.151$, 95% CI = [0.076,0.184], $p < 0.001$), and lower mental health ($b = -0.44$, $f^2 = 0.248$, 95% CI = [0.138,0.279], $p < 0.001$).

**Study and age group effects on financial exploitation measures**
To examine the higher Study 3 OAFEM score compared to Study 1 and Study 2, we first explored within Study 3 to find out whether the younger adults had higher scores than the older adults. We found that younger adults of age below 50 in Study 3 reported higher OAFEM than older adults ($t(634.5) = -11.96$, Cohen's $d = -0.789$, 95% CI = [−0.929, −0.648], $p < 0.001$). After removing younger adults that were uniquely included in Study 3 but not in Study 1 or Study 2, we then explored among only older adults across studies to find out whether there was a study effect on OAFEM scores. Although after excluding younger adults the OAFEM mean of Study 3 decreased from 14.3 to 7.41, there was still a significant study effect on older adults' OAFEM scores across studies (Welch's ANOVA, $F(2, 769.94) = 3.80$, $\eta^2 = 0.010$, 95% CI = [0.000, 0.026], $p = 0.023$). Post-hoc $t$ tests further revealed that Study 3 OAFEM score was significantly higher than Study 2 ($t(695.7) = -2.72$, Cohen's $d = -0.202$, 95% CI = [−0.352, −0.052], $p = 0.020$, corrected using the Benjamini–Hochberg procedure). We did not find any other significant pairwise differences.

To explore the relatively higher FEVS score than OAFEM score within Study 3, we normalized OAFEM and FEVS scores into percentages using each measure's own maximum possible score and conducted repeated-measures ANOVA. Results showed a main effect of measure type ($F(1, 842) = 608.37$, $\eta^2 = 0.419$, 95% CI = [0.372, 0.462], $p < 0.001$), a main effect of age ($F(1, 842) = 190.1$, $\eta^2 = 0.184$, 95% CI = [0.140, 0.229], $p < 0.001$), and an interaction between effects of measure type and age ($F(1, 842) = 10.01$, $\eta^2 = 0.012$, 95% CI = [0.002, 0.030], $p = 0.002$). Follow-up simple main effect analyses show both younger and older adults scored higher on FEVS than on OAFEM, and the difference between the two scores is larger among younger adults than older adults. However, when we conducted the analysis using the full, unfiltered sample (i.e., prior to filtering out participants whose z-transformed OAFEM and FEVS-SF scores differed by an absolute value above two), the interaction was no longer significant ($F(1, 931) = 0.57$, $\eta^2 = 0.001$, 95% CI = [0, 0.008], $p = 0.451$); the main effects of measure type and age remained significant ($F(1, 931) = 379.53$, $\eta^2 = 0.290$, 95% CI = [0.243, 0.334], $p < 0.001$; $F(1, 931) = 242.5$, $\eta^2 = 0.207$, 95% CI = [0.163, 0.250], $p < 0.001$). We also conducted item-wise analyses to explore whether there were age patterns at the item level in OAFEM and FEVS-SF. We found all items in OAFEM and FEVS-SF were negatively associated with age. Younger people thus showed higher vulnerability to exploitation across all items in both OAFEM and FEVS-SF compared to older people.

**Discussion**
In three studies that sampled diversely from older adults of the Philadelphia metropolitan area (Study 1), older adults across Pennsylvania (Study 2), and both younger and older adults across the United States (Study 3), we examined associations between self-reported measures of financial

**Fig. 5 | FEVS-SF interaction.** Scatterplots ($n$ = 844 participants) with regression lines showing associations between FEVS-SF (the Short Form of Financial Exploitation Vulnerability Scale) and the interaction between risk factors. Regression lines represent interaction between everyday cognitive decline and persuadability (**A**) and insensitivity to trustworthiness cues (**B**) with age, race, gender, and education controlled. Different lines represent the relationship between the variables at different levels of the moderator: the dark solid line shows +1 SD (standard deviation) above the mean, the dashed line represents the mean, and the light dashed line indicates −1 SD below the mean of the moderator.

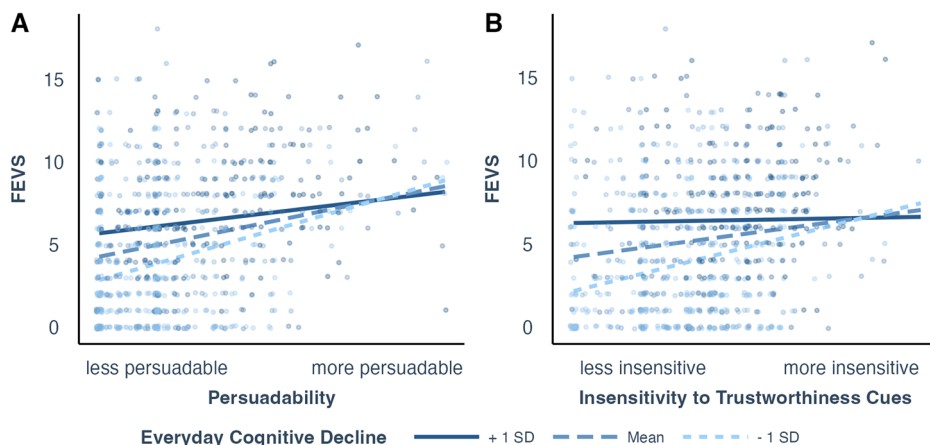

exploitation and the relatively understudied construct of trust in others and fairness preference during economic games, as well as interactions with and between previously-establish putative risk factors. Our goals were to address gaps in the literature of financial exploitation susceptibility in diverse and vulnerable populations, test hypotheses across three separate cohorts, and contribute towards a comprehensive understanding to inform future risk reduction efforts. In Study 1 and Study 2, we found participants with lower socioeconomic status and a higher tendency to deliberately worsen their feelings were at the greatest risk for financial exploitation. Additionally in Study 2, we found participants with lower socioeconomic status and higher self-reported cognitive decline, persuadability, or extrinsic affect-worsening were at the greatest risk for financial exploitation. In Study 3, consistent across two financial exploitation vulnerability measures, we found that higher financial exploitation risk was associated with decreased social support, increased need to belong, increased persuadability, increased insensitivity to trustworthiness cues, and decreased mental health. Across three studies, we consistently found greater insensitivity to trustworthiness cues and persuadability were associated with greater financial exploitation risk. Taken together, our findings suggest that risk for financial exploitation is dependent upon a combination of psychosocial, demographic, and health factors.

One of our primary goals was to investigate how trust in others during economic games may be associated with self-reported financial exploitation. Using the Trust Game (Berg et al.[16]), to measure people's tendency to trust others in financial contexts, we did not find any significant associations between trust and financial exploitation, including interactions with other factors. However, when considering insensitivity to trustworthiness cues and persuadability from the Two-Factor Gullibility Scale (Teunisse et al.[53]), we found consistent statistically significant relationships across all three studies; specifically, greater insensitivity to trustworthiness cues and persuadability were associated with greater financial exploitation risk. Our exploratory analyses following the different results from the Trust Game and the Two-Factor Gullibility Scale showed only weak correlations between the measures (all absolute $rs$ < 0.16, Fig. S1). One possible explanation for the weak correlations might be the weak ecological validity of the Trust Game and Ultimatum Game due to the lack of incentive compatibility. The Trust Games and Ultimatum Game we used to measure trust and fairness preference in each study all used hypothetical scenarios, and the amount of money mentioned in each scenario was relatively small. Moreover, the participant's decisions had no real-world financial consequences, such as bonus money on top of participation compensation. Therefore, participants' responses may not have reflected participants' natural trust and fairness preference in the real world where their related decisions have concrete and often significant financial consequences, such as becoming financial exploitation victims. In contrast, the persuadability and insensitivity to trustworthiness scales ask participants to evaluate their own real-world behavior tendencies. Although there is a chance that participants'

responses might be biased toward showing a more socially and subjectively desirable image (e.g., not easily fooled and sensitive to trustworthiness cues), this possibility only strengthens our findings of significant associations between financial exploitation and persuadability/insensitivity across all three studies.

Previous studies examining the direct effects of SES on risk of financial exploitation have been inconsistent[4,26]. In our research, the main effect of socioeconomic status was also inconsistent across studies. However, when we examined potential interactions between socioeconomic status and other psychosocial factors, one consistent effect we found is the interaction between SES and intrinsic affect-worsening: individuals with lower SES were particularly at risk for financial exploitation when they demonstrated a greater tendency to worsen their own or others' feelings. Future research and interventions aimed at minimizing financial exploitation risk among low SES groups may benefit from further investigating and addressing these interacting factors. Intrinsic affect-worsening (i.e., deliberate worsening of one's feelings) is one such factor worth considering, given multiple observed main and interactive effects. In other words, individuals who were more likely to focus on their own shortcomings, negative experiences or problems were at greater risk for financial exploitation—in general and particularly for lower SES individuals. Intrinsic affect-worsening is associated with emotional exhaustion and health-related impairments[54], and is also observed in clinical depression[57]. Thus, the higher financial exploitation risk we found among lower SES individuals with higher intrinsic affect-worsening suggests depressed individuals from lower SES background are particularly vulnerable to financial exploitation. Such individuals may benefit from targeted interventions focusing on practical strategies to raise awareness of fraud or psychoeducation surrounding the relationship between mood and financial exploitation risk. Future intervention studies should compare the efficacy of community resources geared toward particular at-risk groups versus generic educational resources.

Another potential at-risk group consistently identified from our findings includes individuals with compromised socio-cognitive processing, based on results that poor emotion regulation, insensitivity to trustworthiness cues, self-reported cognitive decline, and higher persuadability were associated with increased financial exploitation risk. Future intervention development may benefit from transferring some socio-cognitive ability-oriented interventions offered in the cognitive remediation literature of acquired brain injuries such as traumatic brain injury or stroke. Social-cognitive abilities, including emotional reasoning and trust evaluations, are often compromised following acquired brain injuries, leading to serious consequences in everyday life, including reduced financial capacity and increased risk for financial exploitation[58–60]. Fortunately, the cognitive rehabilitation literature offers interventions that can help improve emotion recognition and problem solving abilities through cognitive training and training on compensatory strategies[61]. These interventions may be applied to help other vulnerable populations—including subgroups identified in this

study—recognize malicious intent and minimize financial exploitation susceptibility. Example intervention candidates include introducing and training concrete problem solving strategies, internal and external memory strategies that may be used to keep track of prior attempts or negative incidents with certain individuals, and identifying and attending to different types of affective information (e.g., verbal and non-verbal cues)[62]. Future studies may wish to systematically test the feasibility and efficacy of these types of interventions for the purpose of minimizing financial exploitation risk.

Our results also showed some unexpected but nevertheless important patterns of age effect within and across studies. First, within Study 3, younger adults scored higher than older adults on both financial exploitation measures we used (OAFEM and FEVS-SF). This might be the result of younger adults being more likely to report losing money to scams than older adults[35,36]. One possible explanation to more scams reported by younger people is that younger people spend longer time online[63] and are more likely to trust information from social media sites[64], both of which may have increased their exposure to potential scammers.

Another unexpected pattern in our financial exploitation measures is observed when comparing OAFEM with FEVS-SF: both younger and older adults reported higher FEVS-SF than OAFEM. This pattern might be related to the different ways OAFEM and FEVS-SF measure risk for financial exploitation and. OAFEM uses items that describe key *behaviors* related to financial exploitation, whereas FEVS-SF focuses on the *context* in which a person is making a financial decision. People fall prey to financial exploitation only when there are perpetrators to fall prey to. Therefore, people who show higher contextual risk for financial exploitation, as reflected by higher FEVS-SF, may not necessarily have actually had *behavioral* experiences related to financial exploitation. For an extreme example, in a safe environment where there are no perpetrators, even the most at-risk people as measured by FEVS-SF may report zero behaviors related to financial exploitation and score zero on OAFEM. Future research would benefit from examining environmental effects such as neighborhood disadvantage, which have been suggested to be correlated with fraud (e.g., Ranson et al.[65]), on risk for and substantiated cases of financial exploitation.

We also found differences on OAFEM scores across studies. Specifically, Study 3 OAFEM was significantly higher than that of Study 2 even after younger adults are excluded from Study 3. One possible source of such difference is regional difference in financial exploitation related reports. Study 1 and Study 2 collected data from participants living in the Philadelphia Metropolitan Area and Pennsylvania Region, while Study 3 collected data from participants across the nation. There are geographical differences in the prevalence of scams[34]. The increased geographical range of Study 3 may have elevated its financial exploitation score. Another potential explanation to cross-study differences observed in our data is the different time periods during which each study was conducted. We discuss this study effect in the Limitations section.

## Limitations

We note that our study has a few notable limitations which may contribute to the inconsistency within our own results. First, measuring risk for financial exploitation is challenging and often limited to self-report data. Although several different instruments have been developed, it remains unclear how they are associated with risk for specific types of financial exploitation, such as consumer fraud versus imposter schemes[27]. Different types of financial exploitation (e.g., perpetration by a stranger versus a close friend or family member) may be associated with both specific and common risk factors. Others have found that stranger-initiated scams are particularly common among older adults experiencing social isolation[66–70], whereas different mechanisms including lifespan psychosocial and emotional changes posited by the Socioemotional Selectivity Theory may play a role in susceptibility to exploitation by family and friends[14,71,72]. Although we attempted to uncover the role of various risk factors on different aspects of financial exploitation by using two measures (OAFEM and FEVS-SF) that assess different aspects of financial exploitation risk (Study 3), we were unable to investigate the

role of perpetrator type due to limitations of the measures and the survey format of our study. This represents an important future direction for research, and the risk factors identified in the present study should be investigated in the context of specific forms of financial exploitation.

Given the difficulties in collecting and validating people's experiences with financial exploitation for reasons such as scam-related stigma[73], self reports might not be an accurate reflection of actual financial exploitations. In addition, many self-report measures of financial exploitation do not assess frequency of financial exploitation. Without such frequency information, a person who was victimized once by, for example, online survey scam, would look the same as someone who had been victimized by multiple online survey scams, resulting in inaccurate overall amount of financial exploitation experienced. Furthermore, financial exploitation scales that have been in major use are almost developed and validated using older people of 60 years of age or above[74], including the OAFEM and FEVS-SF we used in our own studies. Younger adults' daily activities and experiences can be quite different from older adults and may face different financial exploitation risks. According to reports from the Federal Trade Commission[35,36], younger adults report losing money to scams more often than older adults. The type of scams younger adults report being victims of are often also different from those reported by older adults. For example, younger adults are more likely to report impersonator scams, online shopping scams, job scams, and investment scams, whereas older adults are more likely to report falling victim to tech support scams and prize/sweepstakes/lottery scams. In OAFEM and FEVS-SF, several items of both scales relate to the types of scams that are common among younger people. For example, in OAFEM, items mapping on to "Abuse of Trust" (e.g., "Has someone borrowed money and not paid it back?") and "Theft & Scams" (e.g., "Has someone tricked or pressured you into buying something that you now regret buying? ") may relate to impersonator scams, online shopping scams, and investment scams that more commonly target younger adults. For FEVS–SF, although the items ask about the contextual information of the environment in which a person is making a financial decision rather than specific financial exploitation related behaviors, they assess real-world financial transaction decision-making, with which younger adults may struggle more due to lack of knowledge and/or experience comparing to older adults. Although there are several items in both OAFEM and FEVS-SF that may capture risk for financial exploitation among younger people, formal validation is needed and is beyond the scope of current studies. OAFEM, FEVS-SF, and other financial exploitation scales for older people may not be able to accurately capture financial exploitation risks for younger people. Without a validated financial exploitation scale for both younger and older adults, it will be difficult to have a comprehensive and reliable understanding of financial exploitation, especially its association with age and related factors. All these difficulties in measuring risk for financial exploitation may contribute to the inconsistency in some of the main effects and in interactions between factors that we observed across studies.

Beyond the measurement of financial exploitation—an issue that impacts all studies in this area—we note that our work is also uniquely impacted by world events that took place while the studies were conducted. Specifically, our studies were conducted before the COVID-19 pandemic (Study 1: February, 2020), around a month after the pandemic started (Study 2: April-May, 2020), and around a year and a half into the pandemic (Study 3: July-September, 2021), respectively (Table 1). Financial exploitation may have varied as a function of time relative to the time of the pandemic. According to the American Association of Retired Persons the rate of elder financial exploitation during the pandemic more than doubled compared to pre-pandemic years[75]. This pandemic-related surge of financial exploitation might have contributed to the higher OAFEM score among older adults in Study 3 relative to Studies 1 and 2.

## Conclusion

Our work helps provide a fuller characterization of risk factors that contribute to financial exploitation. Findings across all studies showed trust in others, fairness preference, health-related factors, sociodemographics, and psychosocial factors contributed to a more comprehensive understanding of financial exploitation risk and provide guidance to those fighting against financial exploitation.

## Data availability

All data used in this study are questionnaire and survey data. The data and zip code-based median household income codebook are available on https://doi.org/10.17605/OSF.IO/WXY2V.

## Code availability

The code for analyses is available on https://doi.org/10.17605/OSF.IO/WXY2V.

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

## Acknowledgements

This work was supported by funding from the National Institute on Aging (R01-AG067011 to D.V.S., T32AG066598 to K.H.), a College of Liberal Arts Research Award (to D.V.S.), National Institute of Mental Health (R15-MH122927 to D.S.F.), and Adelphi University (Faculty Development Award to D.S.F.). Publication of this article was funded in part by the Temple University Libraries Open Access Publishing Fund. The funders had no role in study design, data collection and analysis, decision to publish or preparation of the manuscript.

## Author contributions

Y.Y. contributed to formal analysis, visualization, and drafted the original manuscript, as well as participated in review and editing. K.H. contributed to conceptualization and methodology, co-wrote the original draft, and

participated in review and editing. S.K. contributed to conceptualization, performed formal analysis, and participated in review and editing. R.M.L. contributed to formal analysis and review and editing. J.J. and T.G. contributed to conceptualization and review and editing. D.S.F. and D.V.S. contributed to conceptualization and methodology, provided supervision and project administration, participated in review and editing, and secured funding for the project.

## Competing interests

The authors declare no competing interests.
