## [Transparent Peer Review file · Communications Psychology]

Psychological, social, and health-related factors predict risk for financial exploitation

Corresponding Author: Professor David Smith

Version 0:

Decision Letter:

Dear Professor Smith,

Thank you for your patience during the peer-review process. Your manuscript titled "Psychological, social, and health-related factors predict risk for financial exploitation" has now been seen by 3 reviewers, whose comments are appended below. You will see that they find your work of some potential interest. However, they have raised quite substantial concerns that must be addressed. In light of these comments, we cannot accept the manuscript for publication, but would be interested in considering a revised version that fully addresses these serious concerns.

We hope you will find the Reviewers' comments useful as you decide how to proceed. Should additional work allow you to address these criticisms, we would be happy to look at a substantially revised manuscript. If you choose to take up this option, please highlight all changes in the manuscript text file, and provide a detailed point-by-point reply to the reviewers.

Editorially, we consider it important that the revised manuscript provides a better introduction with theoretical and conceptual rationale for the studies and design and ends with the presentation of the hypotheses. We also request that the revised manuscript present all pre-registered hypotheses and associated analyses. Please also provide an extended methods section with additional details on the measures used. We allow unlimited space for methods.

I am attaching a checklist that details critical reporting requirements for the revised manuscript. Please attend to each item and ensure your manuscript is fully compliant. We are requesting that your manuscript aligns with these requirements as this facilitates the evaluation of your manuscript, reducing delays in re-review and potential future acceptance. If your revised manuscript is not aligned with these requests on major issues, such as those concerning statistics, it may be returned to you for further revisions without re-review. Additional information can be found in our style and formatting guide Communications Psychology formatting guide.

If the revision process takes significantly longer than five months, we will be happy to reconsider your paper at a later date, provided it still presents a significant contribution to the literature at that stage.

Please use the following link to submit your

- revised manuscript,
- point-by-point response to the referees' comments,
- cover letter (as a separate document),
- the Editorial Policy Checklist (see below),

- the Reporting Summary (see below), and
- the completed Editorial Request Table (attached):

Link Redacted

Thank you for the opportunity to review your work.

Best regards,

Hannah Hao

Hannah Hao, PhD
Editorial Board Member
Communications Psychology
orcid.org/0000-0002-3342-9132

REVIEWER EXPERTISE:

Reviewer #1 risk factors for financial exploitation
Reviewer #2 risk factors for financial exploitation
Reviewer #3 risk factors for financial exploitation

REVIEWER REPORTS:

Reviewer #1 (Remarks to the Author):

This paper examines psychological, social, and health-related factors as predictors of risk for financial exploitation (FE) across three studies involving online panel surveys. The paper is well-written, ambitious, methodologically sound, and will make a contribution to the FE literature. The results showing multiple risk factors across categories and interactions of risk factors with SES and age are particularly interesting. I do have a few comments and suggestions for improvement.

1. The references to tables in the text of manuscript do not correspond to Table titles in the supplementary materials. Text references tables S4 / S5 for study 1 (should be S1 / S2); S6 / S7 for study 2 (should be S3 / S4); Study 3 should reference S5 / S6 / S7 / S8.

2. The study contains a lot of measures, which in the current version of the manuscript are simply summarized in Table 2 with references and scoring ranges. While there is some description of measures in the text, it would be useful to have additional supplementary materials that show all items and scoring algorithms for each measure to help the reader better interpret the findings.

3. The authors need to explain / justify the use of the zip-code based SES measure in Study 2. Given the central role of SES in the analysis, this is a key design feature. It limits the ability to compare SES effects across studies.

4. The regression models looking at study factors as main effects and testing interactions control only for age. Were other sociodemographic factors like gender, race, education considered as controls, which is typical in many studies of FE? This would strengthen study findings. At minimum, the authors should mention other potential covariates and why they were not included.

5. The main issue I see with the paper is a general lack of focus on the FE construct, including its definition and measurement. The paper uses two FE scales – OAFEM and FEVS – but says little to nothing about what these actually measure. A key distinction in the FE literature is between stranger initiated frauds and scams and FE perpetrated by trusted others. Which of these do the OAFEM and FEVS measure? Many studies are now looking at these sub-types of FE separately, and some different risk factors have been found for each. Note 2 above about providing all items and scoring applies particularly to FE outcome measures used. Also, the paper reports means and standard deviations for the OAFEM and FEVS in the supplemental tables but there is no mention of these and no interpretation is provided. A couple of interesting findings here include higher OAFEM scores in Study 3 (which included all ages) than in studies 1 and 2 which included older adults? Do younger people tend to score higher on the OAFEM? They do score higher on the FEVS. Another finding appears to be that relative / absolute scores are higher on the FEVS (mean ~5 on 0-18 scale) than on the OAFEM (mean ~5 – 14 on 0 -114 scale). Why might this be? This could again be related to age as the FEVS is used only in study 3. In sum, the paper would greatly benefit from a more nuanced discussion and analysis of the FE construct, measurement, and study findings.

Reviewer #2 (Remarks to the Author):

Psychological, social and health related factors predict risk for financial exploitation.

The authors present findings from 3 online survey experiments that assess risk for FE in a large sample. Study 1 and Study 2 examines adults over age 50 only. Study 3 adds a lifespan sample of individuals over age 20. The authors include many diverse measures of individual differences including a novel measure, the role of trust and fairness, as well as many others that have been more commonly used in similar research. The authors choose 2 DVs well validated as tools to assess FE as outcome measures. Overall, the paper replicates some previous findings and explores some novel relationships across the three studies. The area of study is highly significant and understudied from a behavioral economics approach and generally the paper is well written. A few queries below are added to improve the flow and clarify some elements of the manuscript.

1. Please clarify how the term in the abstract cognitive decline is operationalized. I am familiar with the ecog as an informant rated tool that can be used to capture decline- was an informant interviewed as well? Or is there a self report version available? Please clarify if it was modified at all.
2. Similar question for the OAFEM- my experience has been as clinician rated tool, was it modified for use as survey instrument?
3. The focus on constructs such as trust and fairness is novel and adds to the literature. However, in the introduction there is little conceptual development of how specifically these tools may map on to FE. Please describe each tool in the introduction and develop them separately. How would "fairness" ratings relate to specific items on the OAFEM for study 1 and 2 and the OAEFM for study three? Why would you expect sharing of returns to be related to items listed on these tools?
4. For study 3 were there any age effects for FEV or OAFEM? Or patterns in vulnerabilities in older versus younger adults?
5. The authors address the issue that the study only uses FE tools validated for older adults, but broadens sample to include adults over age 20. Please expand this limitation to include a brief discussion of the most common scams /fe seen in younger populations and consider if any items on the OAFEM or FEV would capture them.
6. Please add the actual citations to line 199, 201, 203 etc- there appear to be some placeholders (citation).

Reviewer #3 (Remarks to the Author):

The authors examined trust/fairness preferences and their relationship to FE, as well as how SES and psychosocial/health risk factors interact to predict FE risk. Three separate studies tested their study aims. While examining trust/fairness preferences and FE risk is a novel undertaking, the authors do not provide clear rationale for the other investigations reported in their manuscript and it is unclear what exactly they were aiming to do. The large number of variables examined for interactions with SES lends to confusion regarding their main aim and conclusions. Additionally, the justification for conducting two additional studies is also not clear considering that their main hypotheses were not confirmed in Study 1. Overall, I believe a more formulated theoretical basis for each study/hypothesis test needs to be provided and in general there needs to be more clarity throughout the manuscript regarding study variables and why they were chosen. While I believe this is necessary, I am not sure it can be accomplished in one manuscript given the extremely large number of variables included. Some more specific comments are outlined below.

Introduction:

The link between trust/fairness preferences and financial exploitation needs to be strengthened in the introduction.

In general, the introduction is scant and does not provide enough context and justification for the research questions examined in each study. For example, why was SES a main focus as a moderator? There needs to be more empirical support provided or a strong theoretical basis for the inclusion of SES and other moderating variables. In general, I believe the authors attempted to do too much in this manuscript. The connection between those variables and their first aim/question of examining trust/fairness preferences is unclear. A very large number of psychosocial and health related variables were examined, with very little backing for their inclusion provided in the introduction section. While multiple comparisons were controlled for, there was no specific theoretical justification for inclusion of each variable.

Hypotheses should be outlined in the introduction and not in the methods section, especially considering that they were not modified after results of each of the studies.

It was also unclear to me that trust/fairness preferences were conceptualized as "psychosocial factors" in subsequent study questions. As it read in the introduction, those study questions felt completely separate from the first question regarding how trust/fairness are related to financial exploitation. Consider clarifying this.

Methods

Studies 2 and 3 evenly split recruitment based on certain demographic factors but the study sample was not evenly split on these factors. Did recruitment into these categories not work, or were participants excluded for other reasons?

Why in Study 2 did the authors chose to just use income to represent SES? This does not fully reflect SES. Income may be the more appropriate term to reflect this variable.

More clarification of study measures is needed. What is extrinsic vs. intrinsic affect worsening? Could examples of the trust/fairness questions be provided? The descriptions of the psychosocial factors are very vague. More information is needed for each of these measures.

Several places throughout the manuscript are missing citations (e.g., pg 6 paragraph on Psychosocial Factors).

The rationale for conducting studies 2 and 3 is weak.

Results

Presentation of results is confusing and lack of main effects may warrant a more parsimonious examination of interaction effects. For example, both SES and trust/fairness preferences are not related to the financial exploitation scale. Why would this interaction even be tested? It may be better to first present bivariate associations, then the regressions that include only those psychosocial/health variables that are related to financial exploitation. To this end, a correlation matrix of study measures would be helpful.

The tables are very difficult to read.

Discussion

The conclusion that the trust/fairness measures used may not have been “ecologically valid enough to capture aspects of affective processing and social-economic decision making as accurately as persuadably and insensitivity to trustworthiness scales” cannot be supported by the findings and it is a circular argument.

The paragraph about TBI sees out of the blue and random. More explanation is needed if the authors desire to make this connection.

The authors repeatedly mention the importance of interventions and that their study can inform such development of interventions, but very little detail is provided regarding such interventions. They discuss some when discussing TBI but why would these be useful for/how can these specifically be applied to interventions targeting FE?

EDITORIAL POLICIES

We ask that you ensure your manuscript complies with our editorial policies and reporting requirements.

To that end, we require revised manuscripts to be accompanied by two completed items: a reporting summary that collects information on study design and procedure, and an editorial policy checklist that verifies compliance with all required editorial policies

- <https://www.nature.com/documents/nr-reporting-summary.zip>>Nature Research Reporting Summary
- <https://www.nature.com/documents/nr-editorial-policy-checklist.pdf>>Editorial Policy Checklist

All points on the policy checklist must be addressed. Your revised manuscript can only be sent back to the referees if these checklists are completed and uploaded with the revision.

Notes: If you have submitted a Stage 1 Registered Report, Review, Primer, Comment, or Perspective you do not need to submit these forms. If you have already submitted these forms, you may disregard this request.

* **TRANSPARENT PEER REVIEW:** Communications Psychology uses a transparent peer review system. This means that we publish the editorial decision letters including Reviewers' comments to the authors and the author rebuttal letters online

as a supplementary peer review file. However, on author request, confidential information and data can be removed from the published reviewer reports and rebuttal letters prior to publication. If your manuscript has been previously reviewed at another journal, those Reviewers' comments would not form part of the published peer review file.

Version 1:

Decision Letter:

Dear Professor Smith,

Thank you for your patience during the peer-review process. Your manuscript titled "Psychological, social, and health-related factors predict risk for financial exploitation" has now been seen by 3 reviewers, and I include their comments at the end of this message. They are largely supportive of your revised manuscript. However, upon editorial review, we find that further work is needed before we can make a final decision on the manuscript. In particular, we are requesting a final revision to the manuscript where all preregistered hypotheses and analyses are reported for all studies.

It is our policy that authors must disclose all deviations from the preregistered protocol and explain the rationale for deviation (e.g., flaw, feasibility, suboptimality). In cases of deviation from the preregistered analysis plan for reasons other than fundamental flaw or feasibility, the originally planned analyses must also be reported. You can find our full policy on preregistration here: <https://www.nature.com/commsspsychol/submit/preregistration>

In reviewing your work, we notice that not all preregistered hypotheses are reported, for example in regard to impulsivity and family history of AD/ADRD. We also notice that in some instances there appear to be differences between the reported analyses and the preregistered plan. Some of these have been disclosed. However, unless the reason for the change is a fundamental flaw or feasibility, the originally planned analyses must also be reported. For example, you need to report the results using your original control variables and correction for multiple comparisons in Study 1. In Study 2, it appears you preregistered ANOVAs but report regressions.

Although the additional supplemental tables and in-text disclosure of some analytic deviations are good steps towards complying with our preregistration policy, the requirements are not fully satisfied.

To allow readers to easily ascertain whether hypotheses and analyses were preregistered and conducted as planned, please add a column(s) to your existing tables or add separate tables to allow us and future readers to map your results to your preregistered hypotheses and analysis for each study. Please also add a comprehensive table of all deviations from the preregistration, the reason for the deviation, and where the original planned analysis is reported. To be clear, it is okay to also report the analyses that you think are more appropriate in addition to the preregistered analysis provided the deviation is clearly flagged and the preregistered analysis is included. Only in cases where it is not feasible to run the planned analysis, for example, because it violates the assumptions of the planned statistical test, is it not necessary to report the analysis, but this reasoning must be reported.

We cannot accept your manuscript for publication until it meets these requirements.

Please also note that our statistical reporting requirements require reporting confidence intervals around effect sizes.

We therefore invite you to revise and resubmit your manuscript, along with a point-by-point response. Please highlight all changes in the manuscript text file.

Please use the following link to submit your
- revised manuscript,
- point-by-point response to the referees' comments,
- cover letter (as a separate document),
- the Editorial Policy Checklist (see below), and
- the Reporting Summary (see below):

Link Redacted

Best regards,

Hannah Hao

Hannah Hao, PhD
Editorial Board Member
Communications Psychology
orcid.org/0000-0002-3342-9132

REVIEWER EXPERTISE: risk factors for financial exploitation

REVIEWER REPORTS:

Reviewer #1 (Remarks to the Author):

I think the authors have done an excellent job in presenting detailed responses to all of my concerns.

Reviewer #2 (Remarks to the Author):

Thank you for careful attention to my comments. All of my concerns have been adequately addressed.

Stacey Wood

Reviewer #3 (Remarks to the Author):

The authors have substantially improved the manuscript and their main aims and rationale are much clearer in this new version. I have only some minor comments:

1. Some missing punctuation/typos were found in my reading of the manuscript. For example, pg. 4, line 144 there is a missing period after the parenthesis. Pg 6 lines 223-224, there seems to be a period in place of a comma. Pg. 29, line 971, should read "assess". There may be others that I missed.
2. Re: the example of how specific items on the FEVS-SF relate to fairness preference, the last example (pg. 5, lines 168-171) seems a bit of a stretch. The authors may reconsider using this as an example.
3. The section regarding Qualtrics is confusing (pg 6). I am not aware of an option via Qualtrics that recruits samples and provides payment. Did the authors provide a Qualtrics survey link through a different platform (e.g., prolific)? The authors can ignore this point if I am wrong.
4. Pg. 24, lines 778-780 – what is the meaning of "raw sample"? I am not sure what this analysis is referring to. When did the authors filter by similarity in both measures?
5. Pg. 27, line 894 – "are more likely to trust information from social media sites" – this needs a citation.

EDITORIAL POLICIES

We ask that you ensure your manuscript complies with our editorial policies and reporting requirements.

To that end, we require revised manuscripts to be accompanied by two completed items: a reporting summary that collects information on study design and procedure, and an editorial policy checklist that verifies compliance with all required editorial policies.

- <https://www.nature.com/documents/nr-reporting-summary.zip>>Nature Research Reporting Summary
- <https://www.nature.com/documents/nr-editorial-policy-checklist.pdf>>Editorial Policy Checklist

All points on the policy checklist must be addressed. Your revised manuscript can only be sent back to the referees if these checklists are completed and uploaded with the revision.

Notes: If you have submitted a Stage 1 Registered Report, Review, Primer, Comment, or Perspective you do not need to submit these forms. If you have already submitted these forms, you may disregard this request.

If you experience problems in linking your ORCID, please contact the <http://platformsupport.nature.com/> Platform Support Helpdesk.

Version 2:

Decision Letter:

Dear Professor Smith,

Your manuscript titled "Psychological, social, and health-related factors predict risk for financial exploitation" has now been editorially reviewed, and I am delighted to say that we are happy, in principle, to publish a suitably revised version in Communications Psychology.

We therefore invite you to revise your paper one last time to address a list of editorial requests. At the same time we ask that you edit your manuscript to comply with our format requirements and to maximise the accessibility and therefore the impact of your work.

EDITORIAL REQUESTS:

Please review our specific editorial comments and requests regarding your manuscript in the attached "Editorial Requests Table". Please outline your response to each request in the right hand column. Please upload the completed table with your

manuscript files as a Related Manuscript file.

SUBMISSION INFORMATION:

OPEN ACCESS:

* DATA AVAILABILITY:

Link Redacted

Best regards,

Jennifer Bellingtier

Jennifer Bellingtier, PhD
Senior Editor
Communications Psychology

We thank the editor and reviewers for their time and constructive comments, which helped us improve this manuscript. Based on the reviewer feedback, we have made extensive revisions to our manuscript, which we briefly summarize below:

1) Introduction

We provide more theoretical and conceptual rationale regarding our selection of financial exploitation risk factors, especially trust in others and fairness preferences, as well as our study design. We also now outline hypotheses of each study at the end of the Introduction.

2) Methods

We provide additional details on the measures used, including example items and response options.

3) Additional analyses and results

We added demographic factors of gender, race and education as additional control variables and post-hoc analyses of age related patterns in OAFEM and FEVS within and across studies. All results of updated and additional analyses are reported.

5) Discussion

We have added corresponding discussion of additional results of cross-study and within-study differences regarding OAFEM and FEVS scores. We have also modified our discussion of potential interventions to make it more explicit how our study may inform targeted interventions for vulnerable groups identified from our findings.

6) Figures

We have updated the figures to show raw data points and reflect the updated regression results with additional demographic control variables.

7) Tables

We have broken down and reformatted the tables to make them more readable.

Taken together, we believe these revisions have addressed the comments, suggestions, and concerns that reviewers have raised and have greatly strengthened the manuscript. Below, we provide a point-by-point response to each of the reviewers' feedback.

REVIEWER EXPERTISE:

Reviewer #1 risk factors for financial exploitation

Reviewer #2 risk factors for financial exploitation

Reviewer #3 risk factors for financial exploitation

REVIEWER REPORTS:

Reviewer #1 (Remarks to the Author):

This paper examines psychological, social, and health-related factors as predictors of risk for financial exploitation (FE) across three studies involving online panel surveys. The paper is well-written, ambitious, methodologically sound, and will make a contribution to the FE literature. The results showing multiple risk factors across categories and interactions of risk factors with SES and age are particularly interesting. I do have a few comments and suggestions for improvement.

Response: We thank the reviewer for their positive assessment of the manuscript. We appreciate your comments and suggestions that help strengthen our studies. Below we list point-to-point responses to your feedback and updated contents.

[1.1] The references to tables in the text of manuscript do not correspond to Table titles in the supplementary materials. Text references tables S4 / S5 for study 1 (should be S1 / S2); S6 / S7 for study 2 (should be S3 / S4); Study 3 should reference S5 / S6 / S7 / S8.

Response: We thank the reviewer for pointing out the mismatch between table reference in the text and in the supplementary materials. The supplementary table indices in the main text are corrected to match the supplementary materials.

[1.2] The study contains a lot of measures, which in the current version of the manuscript are simply summarized in Table 2 with references and scoring ranges. While there is some description of measures in the text, it would be useful to have additional supplementary materials that show all items and scoring algorithms for each measure to help the reader better interpret the findings.

Response: We appreciate the reviewer pointing out that information about the items and scoring of the measures we used would help the reader better interpret the finding. We previously uploaded our data, data dictionaries, and scoring codes to OSF. Links to these resources are included in the Data Availability and Code Availability sections of the main text. The links are also provided below for the reviewer's convenience:

Link to data: https://osf.io/wxy2v/?view_only=edf03685a6644ac9aed283df4c73219b.

Link to codes: https://osf.io/wxy2v/?view_only=edf03685a6644ac9aed283df4c73219b

We are hesitant to reproduce contents of established measures in our manuscript. However, we do acknowledge that some more information about each measure would help readers better interpret the findings. To this end, we have added example items and response options to our description of each of the measures we used in the Methods section. Updated contents are listed below.

Financial exploitation measures:

“Financial exploitation To measure financial exploitation, we used the 30-item Older Adult Financial Exploitation Measure (OAFEM, Conrad et al., 2010). OAFEM measures risk for financial exploitation using 30 items that describe key behaviors related to financial exploitation of different severity (A lesser severity item example: “Have there been unexplained disappearances of your money or possessions?” A major severity item example: “Has someone changed the direct deposit destination so as to benefit themselves?”). Each item asks the participant to indicate whether a potential financial exploitation experience occurred at any point during the past twelve months, including the present. Responses were coded using a three-point scale (0 = “No”, 1 = “Suspected”, 2 = “Yes”). The final OAFEM score is a severity-weighted sum across all 30 items (Entitlement Expectation = 1; Lesser Theft & Scams = 2; Major Theft & Scams = 3). The weighted sum of rated items is used as the total score. Total scores range from 0 to 124 and higher scores reflect higher risk for financial exploitation. OAFEM was originally developed as a self-report measure conducted by an interviewer. In our online study we used OAFEM as a pure participant-driven self-report measure and modified the question prompt accordingly (i.e., “Please select an answer for each question. All questions refer to the past 12 months, including the present.”). In addition, due to time constraints and the self-report nature of the survey the subject in each question was referred to in general terms (i.e., “someone”).”

“FEVS-SF is a nine-item scale that evaluates how people make decisions about financial transactions in the real world. Each FEVS-SF item asks about different real-world experiences regarding a participant’s personal finances; responses were coded using a three-point scale wherein response options differed according to the question (e.g. “How confident are you in making big financial decisions?”, 0 = confident; 1 = unsure; 2 = not confident; “How often do you feel anxious about your financial decisions and/or transactions?”, 0 = never or rarely; 1 = sometimes; 2 = often). The final FEVS-SF score is a sum across all nine

items ranging from 0 to 18 and higher scores indicating higher financial exploitation vulnerability.”

Health-related measures:

Physical and Mental Health Summary Scores (Hays et al., 2009). This scale assesses general perceptions of one’s health. The scale consists of four global physical health items on overall physical health (“In general, how would you rate your physical health?”), physical function (“To what extent are you able to carry out your everyday physical activities such as walking, climbing stairs, carrying groceries, or moving a chair?”), pain (“In the past 7 days, how would you rate your pain on average? 0 means ‘no pain’, 10 means ‘worst pain imaginable’), and fatigue (“In the past 7 days, how would you rate your fatigue on average?”), and four global mental health items on quality of life (“In general, how would you rate your quality of life?”), mental health (“In general, how would you rate your mental health, including your mood and your ability to think?”), satisfaction with social activities (“In general, how would you rate your satisfaction with social activities and relationships?”), and emotional problems (“How often have you been bothered by emotional problems?”). Participants provided their answers using a 5-point scale in which higher score indicates better health. All answered items are summed, with total scores ranging from 4 to 20 and higher scores reflecting better health.

ECog:

“Everyday Cognition Scale (Ecog, Farias et al., 2008). This scale is a 12-item global measure of perceived decline of everyday cognitive abilities. The items ask about participants’ performance in different domains of everyday function by asking them to compare how they function now compared to 10 years ago (e.g., “Remembering where you have placed objects.”). Participants can indicate their performance by choosing from a four-point scale (1 = better or no change; 2 = a little worse sometimes; 3 = a little worse all the time; 4 = much worse; there is also a “don’t know” option which was coded as missing data). Ecog was initially developed as an informant-rated tool to measure cognitive decline. Subsequent studies have used it as a self-report measure of subjective cognitive decline and have demonstrated equivalent if not improved performance in predicting mild cognitive decline compared to the informant version (Farias et al., 2017, 2021; see Rabin et al., 2015 for a review). Thus, given the self-reported nature of our survey study, we used the 12-item self-report version of the Ecog as a measure of cognitive decline (e.g., “Please rate your ability to perform certain everyday tasks NOW, as compared to 10 years ago”). The sum of all ratings is divided by

the number of items completed, with total scores ranging from 1 to 4 and higher scores reflecting greater decline.”

Psychosocial measures:

“Need to Belong Scale (Leary et al., 2013). This scale measures participants’ desire for acceptance and belonging by asking how much they agree with specific description of their related behaviors (e.g., “I try hard not to do things that will make other people avoid or reject me.”) Participants indicate how much they agree with the description using a five-point scale (1 = strongly disagree; 2 = moderately disagree; 3 = neither agree nor disagree; 4 = moderately agree; 5 = strongly agree). The sum of all ratings is used as total scores ranging from 5 to 50 and higher scores reflecting higher need to belong.

Perceived Social Support (Zimet et al., 1988). The scale measures participants’ subjective experience of social support by asking how much they agree with specific descriptions of their related experience (e.g, “My family really tries to help me.”). Participants indicate how much they agree with the description using a seven-point scale (1 = very strongly disagree; 2 = strongly disagree; 3 = mildly disagree; 4 = neutral; 5 = mildly agree; 6 = strongly agree; 7 = very strongly agree). The sum of all ratings is divided by the number of items completed, with total scores ranging from 1 to 7 and higher scores reflecting higher perceived social support.

Persuadability and insensitivity to untrustworthiness cues subscales (Teunisse et al., 2020). We used these two subscales from the two-factor Gullibility Scale to measure participants’ propensity to be persuaded and insensitivity to the presence of untrustworthiness cues. Each subscale has six items asking how much the participant agree with specific descriptions of their related behaviors (e.g, “Your family thinks you are an easy target for scammers” from the persuadability subscale; “You are pretty poor at working out if someone is tricking you” from the insensitivity to trustworthiness cues subscale). Participants indicate how much they agree with the description using a seven-point scale (1 = strongly disagree; 2 = disagree; 3 = somewhat disagree; 4 = neither agree nor disagree; 5 = somewhat agree; 6 = agree; 7 = strongly agree). The sum of all ratings is used as total scores of each subscale ranging from 6 to 42 and higher scores reflecting higher persuadability or insensitivity to trustworthiness cues.

Emotional Regulation of Others and Self (Niven et al., 2011). We used this measure to evaluate emotion regulation of improving or worsening the feelings of self (i.e., intrinsic affect) of others (extrinsic affect). For each type of emotion

regulation (i.e., intrinsic affect improving, e.g., “I thought about my positive characteristics to make myself feel better”; intrinsic affect worsening, e.g., “I looked for problems in my current situation to make myself feel worse”; extrinsic affect improving, e.g., “I gave someone helpful advice to try to improve how they felt”; and extrinsic affect worsening, e.g., “I told someone about their shortcomings to try to make them feel worse”), participants are asked to report how much they had used specific strategies try to change their own feelings (intrinsic items) or someone else’s feelings (extrinsic items) (e.g., a extrinsic affect improving item: “I gave someone helpful advice to try to improve how they felt”). Participants answer by choosing from a five-point scale of different amount of effort in using a strategy (1 = not at all; 2 = just a little; 3 = moderate amount; 4 = quite a bit; 5 = a great deal). The sum of all ratings is divided by the number of items completed, with total scores ranging from 1 to 5 and higher scores reflecting greater tendency to regulate the emotion of oneself or others.

“..., the Cognitive Reflection Test (Frederick, 2005; Thomson & Oppenheimer, 2016) which measures the ability or disposition to resist reporting an intuitive response that occurs to one first yet may be wrong (e.g. “How many cubic feet of dirt are there in a hole that is 3 feet deep x 3 feet wide x 3 feet long? Please answer using a number. For example, please type "10" instead of "ten." The correct answer is 0 (zero). An example intuitive answer is 27.)”

[1.3] The authors need to explain / justify the use of the zip-code based SES measure in Study 2. Given the central role of SES in the analysis, this a key design feature. It limits the ability to compare SES effects across studies.

Response: We appreciate the reviewer pointing this out and acknowledge that the use of zip-code based SES in Study 2 limited the ability to compare SES effects across studies. We changed the SES measure to a composite score of two individual SES-related measures we also collected in Study 2, participant’s education and maximum individual annual income. We updated corresponding tables and results in the main text. The significance of results of both main and interaction effects using the updated SES composite score (while controlling for age, gender, and race. More details in our next response regarding control variables) remain the same as when we used zip-code based SES measure. Below is the updated Study 2 result section with changes in estimates and effect size highlighted:

Methods, Study 2 Materials:

“Also, in Study 2 we used a composite score of participant’s education and maximum individual annual income as our measure of socioeconomic status

instead of ZIP code based household income as we planned in our pre-registration. Although this measurement of socioeconomic status deviates from our pre-registration, we believe it facilitates comparisons across studies when examining effects of socioeconomic status.”

Results:

“Similar to Study 1, we again found that OAFEM is positively associated with intrinsic affect-worsening, and this positive association is stronger for individuals of lower socioeconomic status than higher socioeconomic status ($b = -0.57, f^2 = 0.024, p < 0.001$). We also found several other moderating effects of socioeconomic status on the association between extrinsic affect-worsening, everyday cognitive decline, and persuadability (Fig. 2). Compared to individuals with higher socioeconomic status, those with lower socioeconomic status showed stronger positive associations between OAFEM and extrinsic affect-worsening ($b = -0.50, f^2 = 0.053, p < 0.001$), everyday cognitive decline ($b = -0.63, f^2 = 0.008, p < 0.001$), and persuadability ($b = -0.04, f^2 = 0.026, p < 0.001$)”

[1.4] The regression models looking at study factors as main effects and testing interactions control only for age. Were other sociodemographic factors like gender, race, education considered as controls, which is typical in many studies of FE? This would strengthen study findings. At minimum, the authors should mention other potential covariates and why they were not included.

Response: We thank the reviewer for pointing out that we did not include demographic factors like gender, race, and education as controls besides age. We have now added these variables as controls to all analyses where applicable (i.e. when the variables are not or part of the key variables of interest). Adding the control variables did not change the significance of our key results and conclusions. We updated all statistics in the main text and corresponding supplementary tables following changes after adding control variables. Below we list updated Analyses and Results sections with changes in statistics highlighted:

Analysis update in Methods section:

“We included age, gender, race, and education as control variables in all studies when they are not among our primary factors of interest in an analysis (e.g. when an analysis involved socioeconomic status, education was not included as a control variable since education was a component of our socioeconomic status composite score). Although this approach deviates from our pre-registration where we only planned to control for age, we believe it increases rigor and

facilitates comparisons with other studies examining financial exploitation studies (e.g., Hall et al., 2022).”

Results update:

(Study 1)

“Results revealed a significant interaction between socioeconomic status and the intrinsic affect-worsening component of our emotion regulation measures ($b = -0.26$, $r^2 = 0.008$, $p < 0.001$).

...Results from these follow-up simple main effect analyses revealed that, for health-related factors, higher OAFEM was associated with worse physical health ($b = -0.13$, $r^2 = 0.022$, $p < 0.001$), worse mental health ($b = -0.14$, $r^2 = 0.019$, $p < 0.001$), and greater everyday cognitive decline ($b = 0.16$, $r^2 = 0.030$, $p < 0.001$). Among psychosocial factors, higher OAFEM was associated with lower perceived social support ($b = -0.10$, $r^2 = 0.010$, $p < 0.001$) and stronger extrinsic affect-improving ($b = 0.07$, $r^2 = 0.006$, $p < 0.001$), extrinsic affect-worsening ($b = 0.06$, $r^2 = 0.006$, $p < 0.001$), and intrinsic affect-improving ($b = 0.06$, $r^2 = 0.005$, $p < 0.001$). Higher OAFEM was also associated with higher persuadability ($b = 0.17$, $r^2 = 0.036$, $p < 0.001$) and higher insensitivity to trustworthiness cues ($b = 0.10$, $r^2 = 0.011$, $p < 0.001$). No significant association between OAFEM and socioeconomic status was found.”

(Study 2)

“Similar to Study 1, we again found that OAFEM is positively associated with intrinsic affect-worsening, and this positive association is stronger for individuals of lower socioeconomic status than higher socioeconomic status ($b = -0.57$, $r^2 = 0.024$, $p < 0.001$). We also found several other moderating effects of socioeconomic status on the association between extrinsic affect-worsening, everyday cognitive decline, and persuadability (Fig. 2). Compared to individuals with higher socioeconomic status, those with lower socioeconomic status showed stronger positive associations between OAFEM and extrinsic affect-worsening ($b = -0.50$, $r^2 = 0.053$, $p < 0.001$), everyday cognitive decline ($b = -0.63$, $r^2 = 0.008$, $p < 0.001$), and persuadability ($b = -0.04$, $r^2 = 0.026$, $p < 0.001$).

...Similar to Study 1, higher OAFEM scores were associated with worse physical health ($b = -0.13$, $r^2 = 0.017$, $p < 0.001$) and higher insensitivity to trustworthiness cues ($b = 0.07$, $r^2 = 0.005$, $p < 0.001$). Also similar to Study 1, no significant association between OAFEM and socioeconomic status was found.”

(Study 3, OAFEM as FE measure)

“We found higher OAFEM scores were associated with lower perceived social support ($b = -0.09$, $f^2 = 0.011$, $p < 0.001$), an increased need to belong ($b = 0.08$, $f^2 = 0.010$, $p < 0.001$), increased persuadability ($b = 0.21$, $f^2 = 0.057$, $p < 0.001$), and greater insensitivity to trustworthiness cues ($b = 0.14$, $f^2 = 0.035$, $p < 0.001$).

...Compared to younger individuals, older individuals showed weaker positive association between OAFEM and insensitivity to trustworthiness cues ($b = -0.01$, $f^2 = 0.008$, $p = 0.003$) and persuadability ($b = -0.01$, $f^2 = 0.020$, $p = 0.001$). We also found the positive association between OAFEM and persuadability is stronger for people with higher SES than people with lower SES ($b = 0.04$, $f^2 = 0.012$, $p = 0.002$).

...We found that when controlling for age, gender, race, and education, similar to what we found in Study 1 higher OAFEM was associated with worse mental health ($b = -0.10$, $f^2 = 0.015$, $p < 0.001$). We also found, similarly Study 1 and Study 2, higher OAFEM was associated with more everyday cognitive decline ($b = 0.25$, $f^2 = 0.087$, $p < 0.001$) (Table S5).

(Study 3, FEVS-SF as FE measure)

“Again we first examined associations between FEVS-SF and the psychosocial factors of interest after controlling for age, gender, race, and education (Table S7). Similar to OAFEM, higher FEVS was associated with lower perceived social support ($b = -0.20$, $f^2 = 0.047$, $p < 0.001$), higher need to belong ($b = 0.16$, $f^2 = 0.027$, $p < 0.001$), higher persuadability ($b = 0.28$, $f^2 = 0.092$, $p < 0.001$), and higher insensitivity to trustworthiness cues ($b = 0.23$, $f^2 = 0.058$, $p < 0.001$).

...Different from OAFEM, we found the positive associations between FEVS-SF and persuadability and insensitivity to trustworthiness cues were weaker for people reporting higher everyday cognitive decline ($b = -0.12$, $f^2 = 0.019$, $p = 0.001$; $b = -0.19$, $f^2 = 0.016$, $p < 0.001$).

...We found higher FEVS-SF was associated with younger age ($b = -0.39$, $r^2 = 0.183$, $p < 0.001$). We also found higher FEVS-SF was associated with lower socioeconomic status ($b = -0.20$, $f^2 = 0.050$, $p < 0.001$), lower mental health ($b = -0.35$, $f^2 = 0.151$, $p < 0.001$), and lower physical health ($b = -0.44$, $f^2 = 0.248$, $p < 0.001$).

[1.5] The main issue I see with the paper is a general lack of focus on the FE construct, including its definition and measurement. The paper uses two FE scales – OAFEM and FEVS – but says little to nothing about what these actually measure. A key distinction in

the FE literature is between stranger initiated frauds and scams and FE perpetrated by trusted others. Which of these do the OAFEM and FEVS measure? Many studies are now looking at these sub-types of FE separately, and some different risk factors have been found for each. Note 2 above about providing all items and scoring applies particularly to FE outcome measures used.

Response: While we did define financial exploitation as “the illegal or improper use of an adult’s funds or property for another person’s profit or advantage (Conrad et al., 2010)” in line 49-50 at the beginning of the introduction section in our initial manuscript, we thank the reviewer for pointing out that we did not fully explain how we measured FE and did not elaborate on the distinct subtypes of FE based on perpetrator type. To address this, we mention this distinction in the introduction, explain why we chose to use OAFEM and FEVS (FEVS-SF), and discuss our lack of systematically investigating the subtypes of FE as a limitation in the discussion. We also add in the methods section how OAFEM and FEVS measure financial exploitation with example items from each scale. Below we list the updated content with key changes highlighted when they are embedded in relevant texts.

Additions to Introduction regarding perpetrator distinction:

“This gap is particularly important given that about half of financial exploitation cases are perpetrated by strangers (51%) followed by friends and family (34%) (Roberto & Teaster, 2011), and that the mechanisms underlying susceptibility to exploitation by different types of perpetrators may differ (Campbell & Lichtenberg, 2021). These findings would suggest that people’s social information integration regarding trust in others and fairness preferences may play an important role in shaping victims’ economic decisions.

...First, given the focus on older adults in previous financial exploitation research (Ross et al., 2014), in Study 1 we focused on risk factors of financial exploitation within older adults from our local community in Philadelphia, PA, USA. We used the 30-item Older Adult Financial Exploitation Measure (OAFEM, Conrad et al., 2010) that measures key behaviors related to older adults’ financial exploitation. Our primary goals were to identify 1) how trust in others and fairness preference across social context in economic activities may be associated with self-reported financial exploitation, and 2) how the association between individual socioeconomic status and self-reported financial exploitation may be moderated by health-related and psychosocial factors. In Study 2, we sought to replicate our findings from Study 1 in a more diversified population within the Pennsylvania region, USA. Building off findings of interactions between socioeconomic status and several psychosocial factors from Study 1 and Study 2, whose cohorts

included only older people from the Philadelphia metropolitan area and Pennsylvania region and lacked diversity in terms of race (majority White), in Study 3 we expanded our sample's age range, racial diversity, and geographical range, and examined how sociodemographic and health-related factors may moderate the association between psychosocial factors and financial exploitation. Age and geographical location have been related to idiosyncrasies in experiences related to financial exploitation, such as prevalence and types of scams (e.g., Campisi, 2023; Federal Trade Commission, 2022; Federal Trade Commission, 2024). Therefore, given the increased sample diversity in age and geographical range, besides measuring financial exploitation related behaviors using OAFEM, we also employed the Short Form of Financial Exploitation Vulnerability Scale (FEVS-SF, Campbell & Lichtenberg, 2021). FEVS-SF focuses on contextual information of the environment in which a person is making a financial decision and allows us to detect contextual risk of financial exploitation."

Additions to Methods Section for OAFEM:

"OAFEM measures risk for financial exploitation using 30 items that describe key behaviors related to financial exploitation of different severity (A lesser severity item example: "Have there been unexplained disappearances of your money or possessions?" A major severity item example: "Has someone changed the direct deposit destination so as to benefit themselves?"). Each item asks the participant to indicate whether a potential financial exploitation experience occurred at any point during the past twelve months, including the present. Responses were coded using a three-point scale (0 = "No", 1 = "Suspected", 2 = "Yes"). The final OAFEM score is a severity-weighted sum across all 30 items (Entitlement Expectation = 1; Lesser Theft & Scams = 2; Major Theft & Scams = 3). OAFEM was originally developed as a self-report measure conducted by an interviewer. In our online study we used OAFEM as a pure participant-driven self-report measure and modified the question prompt accordingly (i.e., "Please select an answer for each question. All questions refer to the past 12 months, including the present."). In addition, due to time constraints and the self-report nature of the survey, the subject in each question was referred to in general terms (i.e., "someone")."

Additions to Methods Section for FEVS:

"FEVS-SF is a nine-item scale that evaluates how people make decisions about financial transactions in the real world. Each FEVS-SF item asks about different real-world experiences regarding a participant's personal finances; responses

were coded using a three-point scale wherein response options differed according to the question (e.g. “How confident are you in making big financial decisions?”, 0 = confident; 1 = unsure; 2 = not confident; “How often do you feel anxious about your financial decisions and/or transactions?”, 0 = never or rarely; 1 = sometimes; 2 = often). The final FEVS-SF score is a sum across all nine items. Higher total FEVS-SF score indicates higher financial exploitation vulnerability.”

Additions to Discussion Section:

“Different types of financial exploitation (e.g., perpetration by a stranger versus a close friend or family member) may be associated with both specific and common risk factors. Others have found that stranger-initiated scams are particularly common among older adults experiencing social isolation (Alves & Wilson, 2008; DeLiema, 2015; Lachs & Han, 2015; Lee & Geistfeld, 1999; Lichtenberg et al., 2020), whereas different mechanisms including lifespan psychosocial and emotional changes posited by the Socioemotional Selectivity Theory may play a role in susceptibility to exploitation by family and friends (Campbell & Lichtenberg, 2021; Carstensen, 2006; Laumann et al., 2008). Although we attempted to uncover the role of various risk factors on different aspects of financial exploitation by using two measures (OAFEM and FEVS-SF) that assess different aspects of financial exploitation risk (Study 3), we were unable to investigate the role of perpetrator type due to limitations of the measures and the survey format of our study. This represents an important future direction for research, and the risk factors identified in the present study should be investigated in the context of specific forms of financial exploitation.”

Taken together, these revisions help clarify why we chose OAFEM and FEVS as our measure for financial exploitation risk. These revisions also help us introduce the distinction in FE perpetrators and discuss our limitation of lack of systematically investigating the subtypes of FE.

[1.6] Also, the paper reports means and standard deviations for the OAFEM and FEVS in the supplemental tables but there is no mention of these and no interpretation is provided. A couple of interesting findings here include higher OAFEM scores in Study 3 (which included all ages) than in studies 1 and 2 which included older adults? Do younger people tend to score higher on the OAFEM? They do score higher on the FEVS. Another finding appears to be that relative / absolute scores are higher on the FEVS (mean ~5 on 0-18 scale) than on the OAFEM (mean ~5 – 14 on 0 -114 scale). Why might this be? This could again be related to age as the FEVS is used only in

study 3. In sum, the paper would greatly benefit from a more nuanced discussion and analysis of the FE construct, measurement, and study findings.

Response: We thank the reviewer for highlighting that OAFEM score is higher in Study 3 than in studies 1 and 2, and in Study 3 FEVS score is relatively higher than OAFEM. We agree our manuscript can greatly benefit from a more nuanced discussion of these differences in scores across and within study. We added between-study analyses of OAFEM scores and within-study FEVS-OAFEM differences. We have also added discussion of results from these added analyses in the discussion section. Below we list updated contents regarding the added analyses, results, and discussion:

Additional analyses:

“Post-hoc Analyses of Financial Exploitation Measures

One pattern that unexpectedly stood out across three studies is the higher mean of OAFEM in Study 3 (5.4 in Study 1, 4.3 in Study 2, and 14.3 in Study 3). We explored the difference in Study 3 OAFEM scores in two ways. (1) To determine whether higher scores within Study 3 were elevated due to differences in one or the other age sample, we conducted a Welch two-sample t test comparing scores in younger and older adults. (2) We conducted a Welch one-way ANOVA of OAFEM scores among older adults across the three studies to determine if there was a study effect.

Another pattern that unexpectedly stood out from Study 3 is the additional risk measure of FEVS score we included in Study 3 were relatively higher than OAFEM score (FEVS with a mean of 5 on a 0-18 scale vs OAFEM with a mean of 14.25 on a 0 -115 scale) despite that both scores are developed to measure financial exploitation risk. To explore the difference, we normalized FEVS and OAFEM into percentages of each measure’s maximum possible range using each measure’s maximum possible score. We then conducted a repeated measures regression to examine potential effects of measure type (FEVS vs OAFEM) and age (younger adults of age below 50 vs older adults of age 50 or above). We also conducted item-wise analyses to explore item-level patterns of age.”

Additional results:

“Study and Age Group Effects on Financial Exploitation Measures

To examine the higher Study 3 OAFEM score compared to Study 1 and Study 2, we first explored within Study 3 to find out whether the younger adults had higher scores than the older adults. We found that younger adults in Study 3 reported higher OAFEM than older adults ($t(634.5) = -11.96, p < 0.001$). After removing younger adults that were uniquely included in Study 3 but not in Study 1 or Study 2, we then explored among only older adults across studies to find out whether there was a study effect on OAFEM scores. Although after excluding younger adults the OAFEM mean of Study 3 decreased from 14.3 to 7.41, there was still a significant study effect on older adults' OAFEM scores across studies ($F(2, 769.94) = 3.80, p = 0.023$). Post-hoc *t* tests further revealed that Study 3 OAFEM score was significantly higher than Study 2 ($p = 0.020$, corrected using the Benjamini-Hochberg procedure). We did not find any other significant pairwise differences.

To explore the relatively higher FEVS score than OAFEM score within Study 3, we normalized OAFEM and FEVS scores into percentages using each measure's own maximum possible score and conducted repeated-measures ANOVA. Results showed a main effect of measure type ($F(1, 842) = 608.37, p < 0.001$), a main effect of age ($F(1, 842) = 190.1, p < 0.001$), and an interaction between effects of measure type and age ($F(1, 842) = 10.01, p = 0.002$). Follow-up simple main effect analyses show both younger and older adults scored higher on FEVS than on OAFEM, and the difference between the two scores is larger among younger adults than older adults. However, when we used our raw sample that was not filtered by similarity in both measures using z-scores, the interaction was no longer significant ($F(1, 931) = 0.57, p = 0.451$); the main effects of measure type and age remained significant ($F(1, 931) = 379.53, p < 0.001$; $F(1, 931) = 242.5, p < 0.001$). We also conducted item-wise analyses to explore whether there were age patterns at the item level in OAFEM and FEVS-SF. We found all items in OAFEM and FEVS-SF were negatively associated with age. Younger people thus showed higher vulnerability to exploitation across all items in both OAFEM and FEVS-SF compared to older people.”

Additional discussion:

“Our results also showed some unexpected but nevertheless important patterns of age effect within and across studies. First, within Study 3, younger adults scored higher than older adults on both financial exploitation measures we used (OAFEM and FEVS-SF). This might be the result of younger adults being more likely to report losing money to scams than older adults (Federal Trade Commission, 2022, 2024). One possible explanation for more scams reported by younger people is that younger people spend longer time online (Briggs, 2022)

and are more likely to trust information from social media sites, both of which may have increased their exposure to potential scammers.

Another unexpected pattern in our financial exploitation measures is observed when comparing OAFEM with FEVS-SF: both younger and older adults reported higher FEVS-SF than OAFEM. This pattern might be related to the different ways OAFEM and FEVS-SF measure risk for financial exploitation and. OAFEM uses items that describe key *behaviors* related to financial exploitation, whereas FEVS-SF focuses on the *context* in which a person is making a financial decision. People fall prey to financial exploitation only when there are perpetrators to fall prey to. Therefore people who show higher contextual risk for financial exploitation, as reflected by higher FEVS-SF, may not necessarily have actually had *behavioral* experiences related to financial exploitation. For an extreme example, in a safe environment where there are no perpetrators, even the most at-risk people as measured by FEVS-SF may report zero behaviors related to financial exploitation and score zero on OAFEM. Future research would benefit from examining environmental effects such as neighborhood disadvantage, which have been suggested to be correlated with fraud (e.g., Ranson et al., 2019), on risk for and substantiated cases of financial exploitation.

We also found differences on OAFEM scores across studies. Specifically, Study 3 OAFEM was significantly higher than that of Study 2 even after younger adults are excluded from Study 3. One possible source of such difference is regional difference in financial exploitation related reports. Study 1 and Study 2 collected data from participants living in the Philadelphia Metropolitan Area and Pennsylvania Region, while Study 3 collected data from participants across the nation. There are geographical differences in the prevalence of scams (Campisi, 2023). The increased geographical range of Study 3 may have elevated its financial exploitation score. Another potential explanation to cross-study differences observed in our data is the different time periods during which each study was conducted. We discuss this study effect in the Limitations section.”

(Limitations)

“Beyond the measurement of financial exploitation – an issue that impacts all studies in this area – we note that our work is also uniquely impacted by world events that took place while the studies were conducted. Specifically, our studies were conducted before the COVID-19 pandemic (Study 1: February, 2020), around a month after the pandemic started (Study 2: April-May, 2020), and around a year and a half into the pandemic (Study 3: July-September, 2021), respectively (Table 1). Financial exploitation may have varied as a function of

time relative to the time of the pandemic. According to the AARP (American Association of Retired Persons) the rate of elder financial exploitation during the pandemic more than doubled compared to pre-pandemic years (Gunther, 2022). This pandemic-related surge of financial exploitation might have contributed to the higher OAFEM score among older adults in Study 3 relative to Studies 1 and 2.“

Reviewer #2 (Remarks to the Author):

The authors present findings from 3 online survey experiments that assess risk for FE in a large sample. Study 1 and Study 2 examines adults over age 50 only. Study 3 adds a lifespan sample of individuals over age 20. The authors include many diverse measures of individual differences including a novel measure, the role of trust and fairness, as well as many others that have been more commonly used in similar research. The authors choose 2 DVs well validated as tools to assess FE as outcome measures. Overall, the paper replicates some previous findings and explores some novel relationships across the three studies. The area of study is highly significant and understudied from a behavioral economics approach and generally the paper is well written. A few queries below are added to improve the flow and clarify some elements of the manuscript.

Response: We thank the reviewer for their positive assessment of the manuscript. We appreciate your comments that help strengthen our studies. Below we list point-to-point responses to your queries and updated contents.

[2.1] Please clarify how the term in the abstract cognitive decline is operationalized. I am familiar with the Ecog as an informant rated tool that can be used to capture decline- was an informant interviewed as well? Or is there a self report version available? Please clarify if it was modified at all.

Response: We thank the reviewer for asking to clarify how Ecog was operationalized in our studies. As mentioned by the reviewer, both the full and shortened version of Ecog (Farias et al., 2008; Farias et al., 2011) were originally developed as informant-rated tools to measure cognitive decline. Later studies have used both the full and shortened versions as self-report measures of subjective cognitive decline (see Rabin et al., 2015 for a review) and the developers of the original ECog have noted that “a self-report version (used to assess subjective concerns) is as equally or more predictive of development of mild cognitive impairment (MCI) as the informant version” (Farias et al., 2017, 2021). In our studies, we used the shortened 12-item self-report version of the Ecog as a measure of cognitive decline. To clarify how we operationalized cognitive decline and used the Ecog scale as a self-report measure, we have updated the corresponding content in our method section with additional information of Ecog and the prompt we used:

Methods (Study 1, Materials):

“Ecog was initially developed as an informant-rated tool to measure cognitive decline. Subsequent studies have used it as a self-report measure of subjective cognitive decline and have demonstrated equivalent if not improved performance in predicting mild cognitive decline compared to the informant version (Farias et al., 2017, 2021; see Rabin et al., 2015 for a review). Thus, given the self-reported nature of our survey study, we used the 12-item self-report version of the Ecog as a measure of cognitive decline (e.g., “Please rate your ability to perform certain everyday tasks NOW, as compared to 10 years ago”).”

[2.2] Similar question for the OAFEM- my experience has been as clinician rated tool, was it modified for use as survey instrument?

Response: We appreciate the reviewer asking to clarify how we used OAFEM in our studies. The version of OAFEM we used is the short (30 items) version developed by Conrad and colleagues (Self-Report Measure of Financial Exploitation of Older Adults, *The Gerontologist*, 2010). All questions ask the participants about their subjective financial exploitation experiences. The participants completed the self-report questionnaire via interview and did not have to substantiate their answers. In our studies, we used the prompt “someone” as the subject in each question asking about a participant’s financial exploitation experience. Below we list methods section updates to clarify how we used OAFEM:

Methods (Study 1, Materials):

“OAFEM was originally developed as a self-report measure conducted by an interviewer. In our online study we used OAFEM as a pure participant-driven self-report measure and modified the question prompt accordingly (i.e., “Please select an answer for each question. All questions refer to the past 12 months, including the present.”). In addition, due to time constraints and the self-report nature of the survey the subject in each question was referred to in general terms (i.e., “someone”).”

[2.3] The focus on constructs such as trust and fairness is novel and adds to the literature. However, in the introduction there is little conceptual development of how specifically these tools may map on to FE. Please describe each tool in the introduction and develop them separately. How would “fairness” ratings relate to specific items on the OAFEM for study 1 and 2 and the OAEFM for study three? Why would you expect sharing of returns to be related to items listed on these tools?

Response: We appreciate the reviewer pointing out that there is little conceptual development of how specifically trust and fairness preference and corresponding tools we used may map on to FE. We have added conceptual development in the introduction and explain how specific items in OAFEM may relate to trust and fairness preference:

Introduction update 1:

“Ultimatum games assess preferences for fairness by proposing fair and unfair monetary offers to participants and record if or how frequently they accept or reject fair and unfair offers. Fairness preferences have been shown to be associated with contract choice. Employees who are more concerned with fairness tend to choose a bonus contract which offers non-binding bonus payment if their performance is satisfactory (Fehr et al., 2007), which may make people vulnerable to financial exploitation (e.g. accepting a performance-based financial promise that cannot be enforced by a third party and becoming exploited if the performance-based bonus is not fulfilled). The Trust Game assesses trust level by recording the amount of money one decides to invest in another person in expectation of reciprocation. Different levels of trust in others may lead one to invest in a scammer or exploiter, resulting in financial loss. Despite their potential association with financial exploitation risk, the Ultimatum Game and the Trust Game have rarely been studied alongside risk factors for financial exploitation.”

Introduction update 2:

“To quantify the risk for financial exploitation, we employed the Older Adult Financial Exploitation Measure (OAFEM, Conrad et al., 2010, all three studies). The OAFEM is a 30-item scale that measures key behaviors related to older adults’ financial exploitation (Conrad et al., 2010). Each item maps onto one of five financial exploitation concept groups (i.e. Abuse of Trust, Financial Entitlement, Coercion, Signs of Possible Abuse, and Theft & Scams. See Table S9 for item mapping). One of the five concept groups is “Abuse of Trust”, which is assessed with eight items. Several items that are not mapped on “Abuse of Trust” may also relate to trust in others, such as the “Theft & Scam” item “Has someone overcharged you for work or services that were done poorly or never done?” Although OAFEM concept mapping does not have a group directly related to fairness preference, we argue that several items may nevertheless relate to fairness preference. For example, for people who have answered “Yes” to the “Theft & Scam” item “Has someone taken your money to do something for you but never did?”, they may have regarded it was fair to the perpetrator to use

the money to do a certain thing and therefore allowed the perpetrator to take their money.

We also adopted the Short Form of Financial Exploitation Vulnerability Scale (FEVS-SF, Campbell & Lichtenberg, 2021) as an additional measure of risk for financial exploitation (Study 3 only). The FEVS-SF is a nine-item scale that probes contextual features of the environment in which a person is making a financial decision. Unlike the OAFEM, items in the FEVS-SF are not mapped onto specific concept groups by its developers. However, several of the items may still relate to trust and fairness preference (Table S10). For example, people who rated higher on the item “How often do you wish you had someone to talk to about financial decisions, transactions, or plans?” may be more likely to trust financial advice from other people, including unsolicited ones from scammers. For fairness preference, an example item is “How often do you feel downhearted or blue about your financial situation or decisions?”. People who are not sensitive to fairness may be more likely to make financial decisions that are either disadvantageously or advantageously unfair and feel downhearted later on.”

[2.4] For study 3 were there any age effects for FEV or OAFEM? Or patterns in vulnerabilities in older versus younger adults?

Response: We conducted additional post hoc regression analyses of age effects on FEVS and OAFEM, which indicated younger age was associated with higher risk for financial exploitation as measured by FEVS and OAFEM. We also conducted item-wise analyses to explore the patterns in vulnerabilities in older versus younger adults and found younger age was associated with higher risk for financial exploitation across all items in both OAFEM and FEVS. Below we report the results:

“Given that our Study 3 sample included both younger and older adults, we specifically examined a potential main effect of age in both OAFEM and FEVS. We found that, after controlling for gender, race, and education, both higher OAFEM and higher FEVS-SF were associated with younger age ($b = -0.24$, $r^2 = 0.065$, $p < 0.001$; $b = -0.39$, $r^2 = 0.183$, $p < 0.001$). We also conducted item-wise analyses to explore whether there were age patterns at the item level in OAFEM and FEVS-SF. We found all items in OAFEM and FEVS-SF were negatively associated with age. Younger people thus showed higher vulnerability to exploitation across all items in both OAFEM and FEVS-SF compared to older people.”

[2.5] The authors address the issue that the study only uses FE tools validated for older adults, but broadens sample to include adults over age 20. Please expand this limitation to include a brief discussion of the most common scams /fe seen in younger populations and consider if any items on the OAFEM or FEV would capture them.

Response: We thank the reviewer for guiding us to a more thorough discussion of FE and our limitations by including contents regarding FE in younger populations. We add discussion of common scams/fe seen in younger populations and how items on the OAFEM and FEVS would capture them.

Updated discussion:

“According to reports from the Federal Trade Commission (2022, 2024), younger adults report losing money to scams more often than older adults. The type of scams younger adults report being victims of are often also different from those reported by older adults. For example, younger adults are more likely to report impersonator scams, online shopping scams, job scams, and investment scams, whereas older adults are more likely to report falling victim to tech support scams and prize/sweepstakes/lottery scams. In OAFEM and FEVS-SF, several items of both scales relate to the types of scams that are common among younger people. For example, in OAFEM, items mapping on to “Abuse of Trust” (e.g. “Has someone borrowed money and not paid it back?”) and “Theft & Scams” (e.g. “Has someone tricked or pressured you into buying something that you now regret buying?”) may relate to impersonator scams, online shopping scams, and investment scams that more commonly target younger adults. For FEVS–SF, although the items ask about the contextual information of the environment in which a person is making a financial decision rather than specific financial exploitation related behaviors, they assesses real-world financial transaction decision-making, with which younger adults may struggle more due to lack of knowledge and/or experience comparing to older adults. Although there are several items in both OAFEM and FEVS-SF that may capture risk for financial exploitation among younger people, formal validation is needed and is beyond the scope of current studies.”

[2.6] Please add the actual citations to line 199, 201, 203 etc- there appear to be some placeholders (citation).

Response: We apologize for the citation placeholder that should have been replaced by actual citations. All of these errors have been corrected.

Reviewer #3 (Remarks to the Author):

The authors examined trust/fairness preferences and their relationship to FE, as well as how SES and psychosocial/health risk factors interact to predict FE risk. Three separate studies tested their study aims.

While examining trust/fairness preferences and FE risk is a novel undertaking, the authors do not provide clear rationale for the other investigations reported in their manuscript and it is unclear what exactly they were aiming to do. The large number of variables examined for interactions with SES lends to confusion regarding their main aim and conclusions.

Additionally, the justification for conducting two additional studies is also not clear considering that their main hypotheses were not confirmed in Study 1.

Overall, I believe a more formulated theoretical basis for each study/hypothesis test needs to be provided and in general there needs to be more clarity throughout the manuscript regarding study variables and why they were chosen. While I believe this is necessary, I am not sure it can be accomplished in one manuscript given the extremely large number of variables included.

Response: We thank the reviewer for their comments and suggestions that help strengthen our studies. Below we list point-to-point responses to their feedback and updated contents. We agree with the reviewer that we should provide more rationale for examining trust/fairness preference and FE risk. We also agree that more clarity is needed regarding our theoretical basis, main aims and conclusions, including our choices of variables. We have added additional rationale to bolster the justification for Study 2 and 3 given results of Study 1. We have also restructured the manuscript to provide the appropriate context for each study. We believe these revisions fully address the reviewer's points and have strengthened the manuscript.

Some more specific comments are outlined below.

[3.1] Introduction:

The link between trust/fairness preferences and financial exploitation needs to be strengthened in the introduction.

Response: Reviewers 1 and 2 raised a similar concern. We appreciate the current reviewer pointing out as well that the link between trust/fairness preferences and

financial exploitation needs to be strengthened in the introduction, which we believe is an excellent point. We have added conceptual development of the links to the Introduction:

“Ultimatum games assess fairness preference by proposing fair and unfair monetary offers to participants and record if or how frequently they accept or reject fair and unfair offers. Fairness preferences have been shown to be associated with contract choice. Employees who are more concerned with fairness tend to choose a bonus contract which offers non-binding bonus payment if their performance is satisfactory (Fehr et al., 2007), which may make people vulnerable to financial exploitation (e.g. accepting a performance-based financial promise that cannot be enforced by a third party and becoming exploited if the performance-based bonus is not fulfilled). The Trust Game assesses trust level by recording the amount of money one decides to invest in another person in expectation of reciprocation. Different levels of trust in others may lead one to invest in a scammer or exploiter, resulting in financial loss. Despite their potential association with financial exploitation risk, the Ultimatum Game and the Trust Game have rarely been studied alongside risk factors for financial exploitation.

[3.2] In general, the introduction is scant and does not provide enough context and justification for the research questions examined in each study. For example, why was SES a main focus as a moderator? There needs to be more empirical support provided or a strong theoretical basis for the inclusion of SES and other moderating variables.

Response: We appreciate the reviewer’s suggestion to add more context and justification in the introduction. We have updated the Introduction to clarify and justify our research questions and approaches:

“Given that previous work has largely focused on single factors that contribute to risk of financial exploitation, our primary goal was to assess how risk for financial exploitation was moderated by other factors. We specifically focused on how sociodemographic factors (e.g. socioeconomic status, age, etc.) and health-related factors may moderate psychological factors such as psychosocial functionings (e.g., trust in others, fairness preference, persuadability, need to belong, and emotion regulation). Such “sociodemographics vs health”(Study 1 & 2) and “sociodemographic vs psychosocial functioning” (Study 1, 2 & 3) combinations allows for a relatively more comprehensive and holistic approach to examining financial exploitation risk factors, integrating both external (sociodemographic and health-related factors) and internal (psychological)

factors. This approach acknowledges the complex, multifaceted nature of financial exploitation vulnerability.”

[3.3] In general, I believe the authors attempted to do too much in this manuscript. The connection between those variables and their first aim/question of examining trust/fairness preferences is unclear. A very large number of psychosocial and health related variables were examined, with very little backing for their inclusion provided in the introduction section. While multiple comparisons were controlled for, there was no specific theoretical justification for inclusion of each variable.

Response: We thank the reviewer for pushing us towards further articulating our justification for inclusion of the variables. In the introduction we summarized putative individual risk factors that have been identified by previous research as well as inconsistency in their results. However, we do acknowledge that we did not provide enough conceptual development and justification for including potential risk factors of trust in others and fairness preference in financial transactions. We have expanded our introduction to include more conceptual development of inclusion of trust and fairness preference factors using the Ultimatum Game and Trust Game:

“The Ultimatum Game assesses preferences for fairness by proposing fair and unfair monetary offers to participants and measuring their acceptance and rejection rates dependent on offer type. Fairness preferences have been shown to be associated with contract choice. Employees who are more concerned with fairness tend to choose a bonus contract which offers non-binding bonus payment if their performance is satisfactory (Fehr et al., 2007), which may make people vulnerable to financial exploitation (e.g. accepting a performance-based financial promise that cannot be enforced by a third party and becoming exploited if the performance-based bonus is not fulfilled). The Trust Game assesses trust in others by recording the amount of money one decides to invest in another person given subjective expectation of reciprocation. Differing levels of baseline interpersonal trust may relate to individuals’ likelihood of falling prey to bad-faith actors. Despite their potential association with financial exploitation risk, the Ultimatum Game and the Trust Game have rarely been studied alongside risk factors for financial exploitation.”

We also explain why we chose the variables as we clarify and justify our research questions and approaches in our response to the reviewer’s previous comment. We list the response below again for the reviewer’s convenience:

“Given that previous work has largely focused on single factors that contribute to risk of financial exploitation, our primary goal was to assess how risk for financial exploitation was moderated by other factors. We specifically focused on how sociodemographic factors (e.g. socioeconomic status, age, etc.) and health-related factors may moderate psychological factors such as psychosocial functionings (e.g., trust in others, fairness preference, persuadability, need to belong, and emotion regulation). Such “sociodemographics vs health”(Study 1 & 2) and “sociodemographic vs psychosocial functioning” (Study 1, 2 & 3) combinations allows for a relatively more comprehensive and holistic approach to examining financial exploitation risk factors, integrating both external (sociodemographic and health-related factors) and internal (psychological) factors. This approach acknowledges the complex, multifaceted nature of financial exploitation vulnerability.”

[3.4] Hypotheses should be outlined in the introduction and not in the methods section, especially considering that they were not modified after results of each of the studies.

Response: We agree with the reviewer and have moved the hypotheses to introduction:

“We conducted three studies to address the gaps of (1) neglecting to examine trust in others and fairness preference during economic games as financial exploitation risk factors, and (2) neglecting to examine interactions between different risk factors, which might have contributed to inconsistent findings in the literature. In Study 1, given the focus on older adults in previous financial exploitation research (Ross et al., 2014), we focused on risk factors of financial exploitation within older adults from our local community in Philadelphia, PA, USA. Our primary goals were to identify 1) how trust in others and fairness preference across social context in economic activities may be associated with self-reported financial exploitation, and 2) how the association between individual socioeconomic status and self-reported financial exploitation may be moderated by health-related and psychosocial factors. In Study 2, we sought to replicate our findings from Study 1 in a more diversified population within the Pennsylvania region, USA. Building off findings of interactions between socioeconomic status and several psychosocial factors from Study 1 and Study 2, whose cohorts included only older people from the Philadelphia metropolitan area and Pennsylvania region and lacked diversity in terms of race (majority White), in Study 3 we expanded our sample’s age range, racial diversity, and geographical range, and examined how sociodemographic and health-related factors may

moderate the association between psychosocial factors and financial exploitation. “

[3.5] It was also unclear to me that trust/fairness preferences were conceptualized as “psychosocial factors” in subsequent study questions. As it read in the introduction, those study questions felt completely separate from the first question regarding how trust/fairness are related to financial exploitation. Consider clarifying this.

Response: The reviewer raises an excellent point, and we believe our updates of the Introduction in response to several of the reviewer’s comments above include information that would address the current comment. Below we list parts about how trust/fairness relate to financial exploitation and the conceptualization of trust/fairness as “psychosocial factors”:

“The Ultimatum Game assesses preferences for fairness by proposing fair and unfair monetary offers to participants and measuring their acceptance and rejection rates dependent on offer type. Fairness preferences have been shown to be associated with contract choice. Employees who are more concerned with fairness tend to choose a bonus contract which offers non-binding bonus payment if their performance is satisfactory (Fehr et al., 2007), which may make people vulnerable to financial exploitation (e.g. accepting a performance-based financial promise that cannot be enforced by a third party and becoming exploited if the performance-based bonus is not fulfilled). The Trust Game assesses trust in others by recording the amount of money one decides to invest in another person given subjective expectation of reciprocation. Differing levels of baseline interpersonal trust may relate to individuals’ likelihood of falling prey to bad-faith actors. Despite their potential association with financial exploitation risk, the Ultimatum Game and the Trust Game have rarely been studied alongside risk factors for financial exploitation.”

Methods

[3.6] Studies 2 and 3 evenly split recruitment based on certain demographic factors but the study sample was not evenly split on these factors. Did recruitment into these categories not work, or were participants excluded for other reasons?

Response: We aimed at an even split on certain demographic factors and discussed our goals and criteria when setting up our panel studies. The uneven split was resulted from Qualtrics’ failure to achieve the goal, not exclusions after we received the raw data from Qualtrics. We have added to the methods to clarify this discrepancy:

“We aimed to achieve an even split on certain demographic factors and discussed our goals and criteria when setting up each panel study with Qualtrics. However, due to real-life constraints (e.g., difficulty in recruiting non-White racial minorities), Qualtrics was not able to achieve these goals, resulting in an uneven split for some factors.”

[3.7] Why in Study 2 did the authors chose to just use income to represent SES? This does not fully reflect SES. Income may be the more appropriate term to reflect this variable.

Response: We appreciate the reviewer for pointing out this discrepancy and limitation. We revised the SES measure to be a composite score of two individual SES-related measures we also collected in Study 2, participant’s education and maximum individual annual income:

Updates in Methods:

“..., in Study 2 we used a composite score of participant’s education and maximum individual annual income as our measure of socioeconomic status instead of ZIP code based household income as we planned in our pre-registration. Although this measurement of socioeconomic status deviates from our pre-registration, we believe it facilitates comparisons across studies when examining effects of socioeconomic status.”

Updates in Results:

“Similar to Study 1, we again found that OAFEM is positively associated with intrinsic affect-worsening, and this positive association is stronger for individuals of lower socioeconomic status than higher socioeconomic status ($b = -0.57, f^2 = 0.024, p < 0.001$). We also found several other moderating effects of socioeconomic status on the association between extrinsic affect-worsening, everyday cognitive decline, and persuadability (Fig. 2). Compared to individuals with higher socioeconomic status, those with lower socioeconomic status showed stronger positive associations between OAFEM and extrinsic affect-worsening ($b = -0.50, f^2 = 0.053, p < 0.001$), everyday cognitive decline ($b = -0.63, f^2 = 0.008, p < 0.001$), and persuadability ($b = -0.04, f^2 = 0.026, p < 0.001$).”

These revisions add clarity regarding the effects of SES and strengthen our original conclusions. We thank the reviewer for highlighting this limitation.

[3.8] More clarification of study measures is needed. What is extrinsic vs. intrinsic affect worsening? Could examples of the trust/fairness questions be provided? The descriptions of the psychosocial factors are very vague. More information is needed for each of these measures.

Response: We appreciate the reviewer pointing out the need for more clarification of study measures. Below we describe our updates to the Methods section and highlight parts that provide examples of the trust/fairness questions and definition and example items of intrinsic/extrinsic affect worsening:

Financial exploitation measures:

Financial exploitation To measure financial exploitation, we used the 30-item Older Adult Financial Exploitation Measure (OAFEM, Conrad et al., 2010). OAFEM measures risk for financial exploitation using 30 items that describe key behaviors related to financial exploitation of different severity (A lesser severity item example: “Have there been unexplained disappearances of your money or possessions?” A major severity item example: “Has someone changed the direct deposit destination so as to benefit themselves?”). Each item asks the participant to indicate whether a potential financial exploitation experience occurred at any point during the past twelve months, including the present. Responses were coded using a three-point scale (0 = “No”, 1 = “Suspected”, 2 = “Yes”). The final OAFEM score is a severity-weighted sum across all 30 items (Entitlement Expectation = 1; Lesser Theft & Scams = 2; Major Theft & Scams = 3). The weighted sum of rated items is used as the total score. Total scores range from 0 to 124 and higher scores reflect higher risk for financial exploitation. OAFEM was originally developed as a self-report measure conducted by an interviewer. In our online study we used OAFEM as a pure participant-driven self-report measure and modified the question prompt accordingly (i.e., “Please select an answer for each question. All questions refer to the past 12 months, including the present.”). In addition, due to time constraints and the self-report nature of the survey the subject in each question was referred to in general terms (i.e., “someone”).”

Example trust question:

“(e.g. “Consider the following scenario. You are given \$23. You can keep all of this money or share some of it with Charlie, a stranger you haven’t met. Whatever you share with Charlie will be tripled. Importantly, Charlie can keep all of the tripled sum of money to himself or share the tripled sum of money evenly with you. How much would you like to share with Charlie?”).”

Example fairness question:

“(e.g., “Consider the following scenario. Let's imagine Jordan is given \$35, but Jordan has to share some of the \$35 with you. Jordan offers you \$7. If you accept Jordan's offer, you get \$7 and Jordan gets \$28. If you reject Jordan's offer, neither of you get any money. Accept or Reject?”).

Health-related measures:

“*Physical and Mental Health Summary Scores* (Hays et al., 2009). This scale gathers general perception of health. The scale consists of four global physical health items on overall physical health (“In general, how would you rate your physical health?”), physical function (“To what extent are you able to carry out your everyday physical activities such as walking, climbing stairs, carrying groceries, or moving a chair?”), pain (“In the past 7 days, how would you rate your pain on average? 0 means ‘no pain’, 10 means ‘worst pain imaginable’), and fatigue (“In the past 7 days, how would you rate your fatigue on average?”), and four global mental health items on quality of life (“In general, how would you rate your quality of life?”), mental health (“In general, how would you rate your mental health, including your mood and your ability to think?”), satisfaction with social activities (“In general, how would you rate your satisfaction with social activities and relationships?”), and emotional problems (“How often have you been bothered by emotional problems?”). Participants provided their answers using a 5-point scale in which a higher score indicates better health. All answered items are summed, with total scores ranging from 4 to 20 and higher scores reflecting better health.”

“*Everyday Cognition Scale* (Farias et al., 2008). This scale is a 12-item global measure of perceived decline of everyday cognitive abilities. The items ask about participants’ performance in different domains of everyday function by asking them to compare how they function now compared to 10 years ago (e.g., “Remembering where you have placed objects.”). Participants can indicate their performance by choosing from a four-point scale (1 = better or no change; 2 = a little worse sometimes; 3 = a little worse all the time; 4 = much worse; there is also a “don't know” option which was coded as missing data). Ecog was initially developed as an informant-rated tool to measure cognitive decline. Subsequent studies have used it as a self-report measure of subjective cognitive decline and have demonstrated equivalent if not improved performance in predicting mild cognitive decline compared to the informant version (Farias et al., 2017, 2021; see Rabin et al., 2015 for a review). Thus, given the self-reported nature of our survey study, we used the 12-item self-report version of the Ecog as a measure of cognitive decline (e.g., “Please rate your ability to perform certain everyday tasks NOW, as compared to 10 years ago”). The sum of all ratings is divided by

the number of items completed, with total scores ranging from 1 to 4 and higher scores reflecting greater decline.”

Psychosocial measures:

“Need to Belong Scale (Leary et al., 2013). This 10-item scale measures participants’ desire for acceptance and belonging by asking how much they agree with specific description of their related behaviors (e.g., “I try hard not to do things that will make other people avoid or reject me.”) Participants indicate how much they agree with the description using a five-point scale (1 = strongly disagree; 2 = moderately disagree; 3 = neither agree nor disagree; 4 = moderately agree; 5 = strongly agree). The sum of all ratings is used as total scores ranging from 5 to 50 and higher scores reflecting higher need to belong.

Perceived Social Support (Zimet et al., 1988). The 12-item scale measures participants’ subjective experience of social support by asking how much they agree with specific descriptions of their related experience (e.g, “My family really tries to help me.”). Participants indicate how much they agree with the description using a seven-point scale (1 = very strongly disagree; 2 = strongly disagree; 3 = mildly disagree; 4 = neutral; 5 = mildly agree; 6 = strongly agree; 7 = very strongly agree). The sum of all ratings is divided by the number of items completed, with total scores ranging from 1 to 7 and higher scores reflecting higher perceived social support.

Persuadability and insensitivity to untrustworthiness cues subscales (Teunisse et al., 2020). We used these two subscales from the two-factor Gullibility Scale to measure participants’ propensity to be persuaded and insensitivity to the presence of untrustworthiness cues. Each subscale has six items asking how much the participant agree with specific descriptions of their related behaviors (e.g, “Your family thinks you are an easy target for scammers” from the persuadability subscale; “You are pretty poor at working out if someone is tricking you” from the insensitivity to trustworthiness cues subscale). Participants indicate how much they agree with the description using a seven-point scale (1 = strongly disagree; 2 = disagree; 3 = somewhat disagree; 4 = neither agree nor disagree; 5 = somewhat agree; 6 = agree; 7 = strongly agree). The sum of all ratings is used as total scores of each subscale ranging from 6 to 42 and higher scores reflecting higher persuadability or insensitivity to trustworthiness cues.

Emotional Regulation of Others and Self (Niven et al., 2011). We used this measure to evaluate emotion regulation of **improving or worsening the feelings of self (i.e., intrinsic affect) of others (extrinsic affect).** For each type of emotion

regulation (i.e., intrinsic affect improving, e.g., “I thought about my positive characteristics to make myself feel better”; intrinsic affect worsening, e.g., “I looked for problems in my current situation to make myself feel worse”; extrinsic affect improving, e.g., “I gave someone helpful advice to try to improve how they felt”; and extrinsic affect worsening, e.g., “I told someone about their shortcomings to try to make them feel worse”), participants are asked to report how much they had used specific strategies try to **change their own feelings (intrinsic items) or someone else’s feelings (extrinsic items)** (e.g., an extrinsic affect improving item: “I gave someone helpful advice to try to improve how they felt”). Participants answer by choosing from a five-point scale of different amount of effort in using a strategy (1 = not at all; 2 = just a little; 3 = moderate amount; 4 = quite a bit; 5 = a great deal). The sum of all ratings is divided by the number of items completed, with total scores ranging from 1 to 5 and higher scores reflecting greater tendency to regulate the emotion of oneself or others.

“..., the Cognitive Reflection Test (Frederick, 2005; Thomson & Oppenheimer, 2016) which measures the ability or disposition to resist reporting an intuitive response that occurs to one first yet may be wrong (e.g. “How many cubic feet of dirt are there in a hole that is 3 feet deep x 3 feet wide x 3 feet long? Please answer using a number. For example, please type “10” instead of “ten.” The correct answer is 0 (zero). An example intuitive answer is 27.)”

[3.9] Several places throughout the manuscript are missing citations (e.g., pg 6 paragraph on Psychosocial Factors).

Response: We apologize for the citation placeholder that should have been replaced by actual citations. All of these errors have been corrected.

[3.10] The rationale for conducting studies 2 and 3 is weak.

Response: We appreciate the reviewer’s point that the rationale for conducting Study 2 and Study 3 could be strengthened, and we believe the original structure of our manuscript contributed to the lack of clarity. We have significantly restructured the manuscript to ensure there is appropriate justification and rationale for each study. Key revisions are summarized below:

“We conducted three studies to address the gaps of (1) neglecting to examine trust in others and fairness preference during economic games as financial

exploitation risk factors, and (2) neglecting to examine interactions between different risk factors, which might have contributed to inconsistent findings in the literature. First, given the focus on older adults in previous financial exploitation research (Ross et al., 2014), in Study 1 we focused on risk factors of financial exploitation within older adults from our local community in Philadelphia, PA, USA. We used the 30-item Older Adult Financial Exploitation Measure (OAFEM, Conrad et al., 2010) that measures key behaviors related to older adults' financial exploitation. Our primary goals were to identify 1) how trust in others and fairness preference across social context in economic activities may be associated with self-reported financial exploitation, and 2) how the association between individual socioeconomic status and self-reported financial exploitation may be moderated by health-related and psychosocial factors. In Study 2, we sought to replicate our findings from Study 1 in a more diversified population within the Pennsylvania region, USA. Building off findings of interactions between socioeconomic status and several psychosocial factors from Study 1 and Study 2, whose cohorts included only older people from the Philadelphia metropolitan area and Pennsylvania region and lacked diversity in terms of race (majority White), in Study 3 we expanded our sample's age range, racial diversity, and geographical range, and examined how sociodemographic and health-related factors may moderate the association between psychosocial factors and financial exploitation."

We also further justify conducting Study 2 and Study 3 following Study 1 in our response below to the reviewer's related comment on the justification of examining interaction effects when main effects are lacking.

Results

[3.11] Presentation of results is confusing and lack of main effects may warrant a more parsimonious examination of interaction effects. For example, both SES and trust/fairness preferences are not related to the financial exploitation scale. Why would this interaction even be tested? It may be better to first present bivariate associations, then the regressions that include only those psychosocial/health variables that are related to financial exploitation. To this end, a correlation matrix of study measures would be helpful.

Response: Although we appreciate the Reviewer's perspective, we disagree that lack of main effects may warrant a more parsimonious examination of interaction effects. Neither the existence or lack of main effect imply the existence or lack of interaction effect, as noted in many statistical texts (e.g. Cohen, B. (2013). *Explaining Psychological Statistics* (4th Edition). Wiley. Pp. 460-461). For instance, a main effect can appear differently depending on the level of another factor (a.k.a interaction

between factors). Below we provide two diagrams of examples illustrating an interaction effect between Factor A and Factor B when there is no main effect of Factor A (panel A and B). We also provide two diagrams of examples illustrating the lack of an interaction effect between Factor A and Factor B when there is a main effect of Factor A (panel C and D).

Given the focus on interactions (which we hope is clearer in our revised Introduction), it is unclear how much value a correlation matrix would add. Interested readers are welcome to download our data and conduct secondary analyses that speak to different questions more suited to examining bivariate associations between measures.

[3.12] The tables are very difficult to read.

Response: We apologize for the difficulty in reading the tables. Below we list modifications we made to make the tables in both the main manuscript and the supplementary materials easier to read:

(1) broke “Table 1. Study Time and Sample Sociodemographics” in the original manuscript into three separate tables of “Table 1. Study Information and Geographical Range”, “Table 2. Participant Demographics”, and “Table 3. Participant Education Level”, and “Table 4. Participant Personal, Household, and Local Area Economic Information”;

(2) broke original “Table 2. Scale List of Financial Exploitation, Sociodemographics, and Psychosocial Factors” into three separate tables of “Table 5. Financial Exploitation Measures”, “Table 6. Health-related Measures”, and “Table 7. Psychosocial Measures”;

(3) revised table layout;

(4) changed the orientation of table pages from portrait to landscape;

(5) added page breaks to allow each table to appear on a page in its entirety as much as it can.

Discussion

[3.13] The conclusion that the trust/fairness measures used may not have been “ecologically valid enough to capture aspects of affective processing and social-economic decision making as accurately as persuadability and insensitivity to trustworthiness scales” cannot be supported by the findings and it is a circular argument.

Response: The reviewer raises an interesting point about potential circularity in our reasoning, and we regret that our initial submission was unclear in these respects. To make our argument more explicit and clear, we added example items of persuadability and insensitivity to trustworthiness scales in Methods section and expanded related contents in Discussion section:

In Methods:

“Persuadability and insensitivity to untrustworthiness cues subscales (Teunisse et al., 2020) We used these two subscales from the two-factor Gullibility Scale to measure participants’ propensity to be persuaded and insensitivity to the presence of untrustworthiness cues. Each subscale has six items asking how much the participant agree with specific descriptions of their related behaviors (e.g, “Your family thinks you are an easy target for scammers” from the persuadability subscale; “You are pretty poor at working out if someone is tricking

you” from the insensitivity to trustworthiness cues subscale). Participants indicate how much they agree with the description using a seven-point scale (1 = strongly disagree; 2 = disagree; 3 = somewhat disagree; 4 = neither agree nor disagree; 5 = somewhat agree; 6 = agree; 7 = strongly agree). The sum of all ratings is used as total scores of each subscale ranging from 6 to 42 and higher scores reflecting higher persuadability or insensitivity to trustworthiness cues.”

In Discussion:

“One possible explanation to the weak correlations might be the weak ecological validity of the Trust Game and Ultimatum Game due to the lack of incentive compatibility. The Trust Game and Ultimatum Game we used to measure trust and fairness preference in each study all used hypothetical scenarios, and the amount of money mentioned in each scenario was relatively small. Moreover, the participant’s decisions had no real-world financial consequences, such as bonus money on top of participation compensation. Therefore participants’ responses may not have reflected participants’ natural trust and fairness preferences in the real world where their related decisions have concrete and often significant financial consequences, such as becoming financial exploitation victims. In contrast, the persuadability and insensitivity to trustworthiness scales ask participants to evaluate their own real-world behavior tendencies. Although there is a chance that participants’ responses might be biased toward showing a more socially and subjectively desirable image (e.g. not easily fooled and sensitive to trustworthiness cues), this possibility only strengthens our findings of significant associations between financial exploitation and persuadability/insensitivity across all three studies.”

[3.14] The paragraph about TBI sees out of the blue and random. More explanation is needed if the authors desire to make this connection.

Response: We thank the reviewer pointing out this issue and we have made the following revisions to improve the flow of the writing while giving more explanation for the point:

“Another potential at-risk group consistently identified from our findings includes individuals with compromised socio-cognitive processing, based on results that poor emotion regulation, insensitivity to trustworthiness cues, self-reported cognitive decline, and higher persuadability were associated with increased financial exploitation risk. Future intervention development may benefit from transferring some socio-cognitive ability-oriented interventions offered in the cognitive remediation literature of acquired brain injuries such as traumatic brain

injury or stroke. Social-cognitive abilities, including emotional reasoning and trust evaluations, are often compromised following acquired brain injuries, leading to serious consequences in everyday life, including reduced financial capacity and increased risk for financial exploitation (Martin et al., 2012; Sunderaraman et al., 2019; Xiao et al., 2017). Fortunately, the cognitive rehabilitation literature offers interventions that can help improve emotion recognition and problem solving abilities through cognitive training and training on compensatory strategies (e.g. Babbage, 2014). These interventions may be applied to help other vulnerable populations - including subgroups identified in this study - recognize malicious intent and minimize financial exploitation susceptibility. Example intervention candidates include introducing and training concrete problem solving strategies, internal and external memory strategies that may be used to keep track of prior attempts or negative incidents with certain individuals, and identifying and attending to different types of affective information (e.g., verbal and non-verbal cues) (Cicerone et al., 2019). Future studies may wish to systematically test the feasibility and efficacy of these types of interventions for the purpose of minimizing financial exploitation risk.”

[3.15] The authors repeatedly mention the importance of interventions and that their study can inform such development of interventions, but very little detail is provided regarding such interventions. They discuss some when discussing TBI but why would these be useful for/how can these specifically be applied to interventions targeting FE?

Response: We thank the reviewer for this comment and for encouraging us to be more clear about intervention implications. We aimed to provide some new directions based on our findings while remaining within the boundaries of our competence. We have made edits and additions to the text to make the discussion of our findings’ implication on potential intervention more explicit and specific:

“...Intrinsic affect-worsening is associated with emotional exhaustion and health-related impairments (Niven et al., 2011), and is also observed in clinical depression (Gotlib & Joormann, 2010). Thus, the higher financial exploitation risk we found among lower SES individuals with higher intrinsic affect-worsening suggests depressed individuals from lower SES background are particularly vulnerable to financial exploitation. Such individuals may benefit from targeted interventions focusing on practical strategies to raise awareness of fraud or psychoeducation surrounding the relationship between mood and financial exploitation risk. Future intervention studies should compare the efficacy of community resources geared toward particular at-risk groups versus generic educational resources.

Another potential at-risk group consistently identified from our findings includes individuals with compromised socio-cognitive processing. Specifically, those with poor emotion regulation, insensitivity to trustworthiness cues, and higher persuadability were associated with increased financial exploitation risk. Future intervention development may benefit from transferring some socio-cognitive ability-oriented interventions offered in the traumatic brain injury (TBI) cognitive remediation literature. Social-cognitive abilities, including emotional reasoning and trust evaluations, are often compromised following traumatic brain injury (TBI), leading to serious consequences in everyday life, including reduced financial capacity and increased risk for financial exploitation (Martin et al., 2012; Sunderaraman et al., 2019; Xiao et al., 2017). Fortunately, the TBI cognitive remediation literature offers interventions that can help improve emotion recognition and problem solving abilities through cognitive rehabilitation post-TBI (Babbage, 2014). These interventions may be translated to help other vulnerable populations - including subgroups identified in this study - recognize malicious intent and minimize financial exploitation susceptibility. Example intervention candidates include introducing and training concrete problem solving strategies, internal and external memory strategies that may be used to keep track of prior attempts or negative incidents with certain individuals, and identifying and attending to different types of affective information (e.g., verbal and non-verbal cues) (Cicerone et al., 2019). Future studies may wish to systematically test the feasibility and efficacy of these types of interventions for the purpose of minimizing financial exploitation risk.”

Reviewer #1 (Remarks to the Author):

I think the authors have done an excellent job in presenting detailed responses to all of my concerns.

Response: We thank the reviewer for affirming the changes we made. We greatly appreciate their valuable input, which has significantly helped improve the quality of our manuscript.

Reviewer #2 (Remarks to the Author):

Thank you for careful attention to my comments. All of my concerns have been adequately addressed.

Stacey Wood

Response: We thank the reviewer for acknowledging our efforts and affirming the changes we made. We greatly appreciate their valuable input, which has significantly helped strengthen our manuscript.

Reviewer #3 (Remarks to the Author):

The authors have substantially improved the manuscript and their main aims and rationale are much clearer in this new version. I have only some minor comments:

Response: We thank the reviewer for acknowledging our efforts and affirming the changes we made. We greatly appreciate their valuable input, which has significantly helped strengthen the aims and rationale of our manuscript. Below we list point-to-point responses to their further minor comments.

1. Some missing punctuation/typos were found in my reading of the manuscript. For example, pg. 4, line 144 there is a missing period after the parenthesis. Pg 6 lines 223-224, there seems to be a period in place of a comma. Pg. 29, line 971, should read “assess”. There may be others that I missed.

Response: We appreciate the reviewer for pointing out the punctuation and typographical errors and appreciate their attention to detail. We reviewed and corrected all identified typos during proofreading.

2. Re: the example of how specific items on the FEVS-SF relate to fairness preference, the last example (pg. 5, lines 168-171) seems a bit of a stretch. The authors may reconsider using this as an example.

Response: We thank the reviewer for their thoughtful feedback regarding the example FEVS-SF item provided for fairness preference. We appreciate your observation and have reconsidered the example. We have changed the example to a more relevant one and explained how it may relate to fairness preference.

Updated text:

“For fairness preference, people who rated higher on the item “How often do you feel anxious about your financial decisions and/or transactions?” may be anxious about being treated unfairly when making financial decisions and fall prey to scammers who offer ostensibly fair deals or ostensibly unfair deals that are allegedly more advantageous to the victim.”

3. The section regarding Qualtrics is confusing (pg 6). I am not aware of an option via Qualtrics that recruits samples and provides payment. Did the authors provide a Qualtrics survey link through a different platform (e.g., prolific)? The authors can ignore this point if I am wrong.

Response: We thank the reviewer for their question regarding the use of Qualtrics for participant recruitment and compensation. In our study, we utilized Qualtrics Panel Services. Qualtrics Panel Services handled both recruitment and payment, similar to other large-scale survey platforms. Other studies have also used Qualtrics Panel Services for both participant recruitment and compensation. For example, in a study that compared different online survey recruitment platforms (Ibarra et al., 2018; full bibliography provided below), the participants were recruited and paid through Qualtrics Panel Services (described in the “Qualtrics Panel” subsection in the Recruitment Platforms section)

Ibarra, J. L., Agas, J. M., Lee, M., Pan, J. L., & Bottenheim, A. M. (2018). Comparison of Online Survey Recruitment Platforms for Hard-to-Reach Pregnant Smoking Populations: Feasibility Study. *JMIR Research Protocols*, 7(4), e101.
<https://doi.org/10.2196/resprot.8071>

4. Pg. 24, lines 778-780 – what is the meaning of “raw sample”? I am not sure what this analysis is referring to. When did the authors filter by similarity in both measures?

Response: We thank the reviewer for pointing out this area of confusion. We apologize for the lack of clarity in our terminology and explanation. We have revised this section to provide a clearer description.

Updated text:

“However, when we conducted the analysis using the full, unfiltered sample (i.e., prior to filtering out participants whose z-transformed OAFEM and FEVS-SF scores differed by an absolute value above two), the interaction was no longer significant.”

5. Pg. 27, line 894 – “are more likely to trust information from social media sites” – this needs a citation.

Response: We thank the reviewer for pointing out the missing citation. We added the citation and full reference.

Updated text:

“...are more likely to trust information from social media sites (Liedke & Gottfried, 2022)...”

Full reference:

“Liedke, J., & Gottfried, J. (2022, November 4). *Trust in social media is changing. Here’s how it breaks down by age*. World Economic Forum.
<https://www.weforum.org/stories/2022/11/social-media-adults-information-news-platforms/>